# StreetDiffusion: Street Scenes Generation via Multi-view Stable Diffusion with Structure Prompts

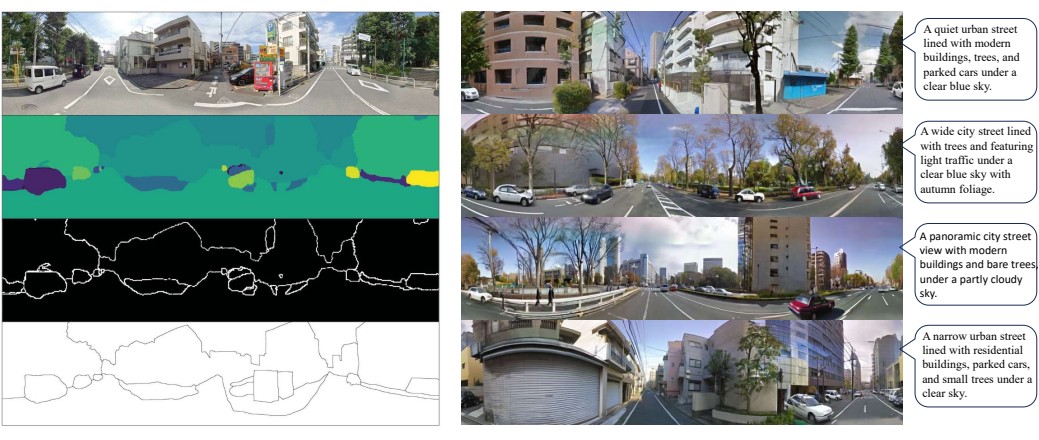

Figure 1: The left side of the figure illustrates representative sample and structure prompts from our proposed Street360 dataset, while the right side presents representative generated results. Note that the examples shown on the left and right are not paired; they are independently selected to illustrate the dataset characteristics and the model's generation capability, respectively.

## Abstract

Multi-view Stable Diffusion has been proposed and applied for indoor or wild scene generation. However, the generation of outdoor scenes, especially urban street scenes, has not yet been well studied, which is more complicated than existing indoor or wild scene generation due to the fact that it contains more objects and structures. In this work, we focus on the generation of street scenes relying on a multi-view stable diffusion model with structure prompts, such as segmentation maps, contour maps, or user sketches. Thus, we propose StreetDiffusion, which employs a Panorama–Perspective Synergy Framework to integrate panoramic and local information, where structural priors are inserted into two branches to generate highly consistent and realistic multi-view street scene images. To study the street scene generation issue, we propose a large multi-view street scene dataset, Street 360, consisting of 10K multi-view and panorama images from urban streets. Experiments demonstrate that the proposed StreetDiffusion model generates high-quality street scenes, with a clear advantage on the street scene generation task over existing multi-view generation models designed for indoor or wild scenes.

## 1 Introduction

Text-based multi-view image generation (Ramesh et al., 2021; Nichol et al., 2022; Ramesh et al., 2022; Rombach et al., 2022; Saharia et al., 2022) has emerged as a promising direction in computer vision with applications in VR(Yang et al., 2023), AR, video games, and film production. Early

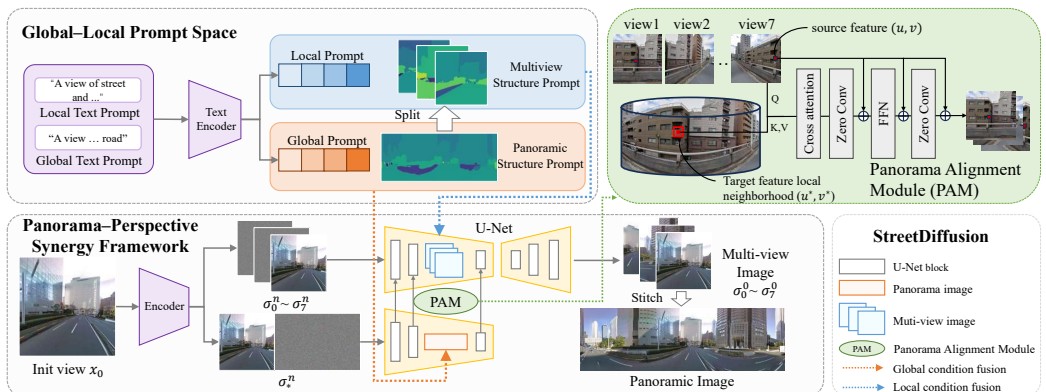

Figure 2: The pipeline of StreetDiffusion. The top is the Global-Local Prompt Space, including local and global text prompts, and the corresponding structure prompts. The global text is composed by concatenating the local texts, while the multi-view structure prompt is derived by segmenting the panoramic structure prompt. The bottom is a Panorama–Perspective Synergy Framework: a panoramic generation and a multi-view generation branch. The global and local conditions are inserted into the panorama and multi-view generation branches, respectively.

methods (Wang et al., 2023a; Bar-Tal et al., 2023; Lee et al., 2023; Liu et al., 2025) extended pre-trained diffusion models to panoramic generation by iteratively synthesizing local views, but were mainly constrained to indoor scenes. Later approaches (Zhang et al., 2024b; Chen et al., 2022; Tang et al., 2023) fine-tuned diffusion models to directly generate 2:1 panoramas, yet their resolution remained limited by the original architectures (e.g., 1024×512). As a result, outdoor panoramas, particularly street scenes, remain challenging due to their complex layouts and dynamic elements.

In summary, two challenges hinder progress: (*i*) most existing datasets (Dai et al., 2017; Chang et al., 2017; Chen et al., 2022; Xiao et al., 2012) emphasize indoor or overly simple natural scenes, lacking urban complexity; (*ii*) current methods often produce artifacts such as distorted buildings and warped road structures when applied to street scenes.

To address these issues, we introduce **Street360**, a large-scale dataset with over 10k real-world multi-view panoramas of urban streets, enriched with text prompts and structural annotations (Fig. 1). Building on this, we propose **StreetDiffusion**, a panorama–perspective synergy framework based on multi-view diffusion. Our key idea is to integrate structural information into both the panoramic and multi-view branches—serving as global cues in the former and local guidance in the latter—while enabling their interaction via attention-based coordinate projection (Fig. 2). This design significantly improves realism, consistency, and controllability in street panorama generation. In summary, the contributions of the paper are as follows.

- As far as we know, this is the first work on the street scene generation full of various object structures. Most existing methods focus on simple indoor or wild scenes. Besides, we propose a large new dataset, Street360, specially designed for studying the issue.
- We propose a novel Panorama–Perspective Synergy Framework, **StreetDiffusion**, which effectively leverages structure prompts (*e.g.*, segmentation maps, contour maps, or user sketches) to guide the generation, ultimately producing street images that accurately adhere to the given structural cues. Furthermore, to exploit global contextual information and enforce consistency across multiple views, we introduce the **Panorama Alignment Module (PAM)**, which establishes geometry-aware correspondences between panoramic and perspective representations.
- Extensive experimental results demonstrate our method achieves the best performance in street scene generation, surpassing previous models in terms of generation quality and multi-view consistency.

## 2 RELATED WORK

**Pantropic Image Generation.** Several methods (Wang et al., 2023b; Dastjerdi et al., 2022; Akimoto et al., 2022; Chen et al., 2022; Zhang et al., 2024b; Wang et al., 2022; Oh et al., 2022; Wang et al.,

2024; Sun et al., 2025; Zheng et al., 2025; Yang et al., 2024) have been proposed for panoramic scene generation. These works either progressively outpaint panoramas in an autoregressive way, or generate the entire panorama in one shot . Other methods, such as MultiDiffusion (Bar-Tal et al., 2023) and SyncDiffusion (Lee et al., 2023), divide the panorama into multiple regions, generating them step by step to address inconsistencies between regions. PanFusion (Zhang et al., 2024b) takes this step further by integrating a panoramic branch and a multi-view branch, leveraging their projection relationships to achieve more realistic panorama generation, but the generated images are of relatively low resolution. OPa-Ma (Gao et al., 2024) uses lightweight Mamba to model long-range dependencies in panoramas and leverages text for better semantic consistency. *However, while these methods have demonstrated promising results in indoor panorama generation, but they struggle to maintain realism and coherence when applied to outdoor scenes, highlighting the need for further improvements.*

**Multi-view Generation.** While single-view image generation (Ho et al., 2020; Dhariwal and Nichol, 2021; Saharia et al., 2022) has become relatively mature, multi-view image generation still faces several challenges. For example, MVDiffusion (Tang et al., 2023) maintains consistency between adjacent views using the Correspondence-Aware Attention (CAA) module; however, issues such as duplicated objects and repetitive textures may still arise. DiffCollage (Zhang et al., 2023b) decomposes large images into factor graphs and generates them with multiple diffusion models in parallel, but struggles with tasks requiring long-range consistency, often producing artifacts such as duplicated structures (e.g., repeated snake tails). In the field of 3D generation, some methods (Liu et al., 2024; 2023; Tang et al., 2025; Metzer et al., 2023), such as Zero-1-to-3 (Liu et al., 2023), leverage geometric relationships across multiple views to predict a complete 3D representation, while Latent-NeRF (Metzer et al., 2023) integrates NeRF as a 3D prior and applies diffusion in the latent space, resulting in more natural and consistent multi-view generation. These approaches provide valuable insights into multi-view diffusion models, yet improving view consistency and geometric accuracy remains an open challenge. *To address this, we propose a Panorama–Perspective Synergy Framework based on structural information. The structural information enhances the geometric stability of the generated images, while the Panorama–Perspective Synergy Framework integrates global and local information to further improve consistency, effectively mitigating instability and repetitive texture issues in multi-view generation.*

**Outdoor Generation.** Although existing methods (Chen et al., 2022; Akimoto et al., 2022; Bar-Tal et al., 2023; Lee et al., 2023; Tang et al., 2023; Zhang et al., 2024b; Liu et al., 2025) have achieved promising results in indoor scene generation, the complexity of outdoor scenes often makes their generation challenging. In practical applications, such as autonomous driving, it is crucial to generate realistic and coherent outdoor multi-view images. Currently, DrivingDiffusion (Li et al., 2024) and PERLDIFF(Zhang et al., 2024a) leverage 3D semantic boxes for generation, while PanoFree (Liu et al., 2025) and MVDiffusion (Tang et al., 2023) can generate certain outdoor scenes, but their outputs remain relatively simplistic, often limited to single grasslands, deserts, or basic city parks, failing to capture more complex street environments. Mixed-View Panorama Synthesis using Geospatially Guided Diffusion (Xiong et al., 2024) further explores synthesizing target panoramas by combining satellite imagery and nearby street-level views with geospatial attention. In addition, recent works such as CrossViewDiff (Chen et al., 2024) and SkyDiffusion (Ye et al., 2024) leverage BEV (Bird's-Eye View) representations in diffusion models to bridge large viewpoint gaps via structure-texture control and curved-BEV / multi-to-one mapping strategies. Streetscapes(Deng et al., 2024) is capable of generating high-quality single-view street scene videos. StreetCrafter (Yan et al., 2025) uses LiDAR-rendered point clouds as pixel-level input in a controllable video diffusion model to synthesize street sequences with precise camera control. Controllable Satellite-to-Street-View Synthesis (Ze et al., 2025) applies an Iterative Homography Adjustment during the diffusion to maintain accurate pose alignment and supports zero-shot control of lighting and weather conditions. Geometry-Guided Cross-View Diffusion (Lin et al., 2025) introduces a geometry-guided condition (GCC) to explicitly model the one-to-many correspondence between satellite and street views, resolving geometric ambiguity.

Compared to these approaches, this paper focuses on generating urban street scenes, achieving detailed and realistic renderings of key elements such as buildings, vehicles, and crosswalks. *To this end, our proposed StreetDiffusion, a Panorama–Perspective Synergy Framework combining panoramic and multi-view generation, introduces structure guidance to provide a comprehensive solution for text-driven multi-view image and 360 panorama generation.*

# 3 MULTI-VIEW STREET SCENE GENERATION

The goal of this work is to generate a coherent panoramic image, aligned with user intent, restricted to non-polar regions to avoid distortions, given a single target view and its corresponding structural prompts. The structural prompts can take various forms, such as contour maps, semantic maps, or user-drawn sketches. As illustrated in Fig. 2, our **Global–Local Prompt Space** extracts both local and global textual descriptions as well as contour-based structural prompts (Sec. 3.1). These serve as inputs to the **Panorama–Perspective Synergy Framework**, which generates eight perspective views that are subsequently stitched into a complete panorama (Sec. 3.2). To effectively leverage global information, we propose the **Panorama Alignment Module (PAM)**, which maps each point from a perspective view to a local neighborhood in the panorama, thereby ensuring cross-view consistency (Sec. 3.3). Finally, to further enhance the quality of the **Panorama–Perspective Synergy Framework**, we conduct separate training for the panoramic and perspective branches, followed by joint training with PAM (Sec. 3.4).

## 3.1 GLOBAL–LOCAL PROMPT SPACE

As the first stage of our pipeline, we construct a unified prompt space that bridges panoramic images and their corresponding multi-view representations. This prompt space integrates both **textual** and **structural** conditions, capturing scene semantics at different levels of granularity.

**Text Prompts.** StreetDiffusion combines local and global text prompts. Specifically, we first realize (Wang, 2025) a panorama by generating eight perspective views, each possessing a horizontal field of view of 90° with a 45° overlap and employ BLIP-3 (Xue et al., 2024) to generate initial local text prompts for each view. However, adjacent views often yield highly similar descriptions, which limits their discriminative power. To address this, we introduce GPT-4 to regenerate more detailed local prompts for each perspective view by leveraging both the panoramic image and the BLIP-3 outputs. Finally, all refined local prompts are concatenated to form a global text prompt, providing a more comprehensive and semantically rich scene description. All refined local prompts are used as the multi-view text inputs for MVDiffusion and for the multi-view branch of our method, while the concatenated global prompt serves as the text input for panoramic baselines and for the panoramic branch of our model. The semantic information contained in both forms is identical, and the distinction lies only in whether the prompts are supplied separately or as a single merged description.

**Structure Prompts.** We use three types of structure prompts (see Fig. 1) in the StreetDiffusion model: segmentation maps, contour maps, and user-input sketches. Panoramic segmentation maps are obtained using OneFormer (Jain et al., 2023), and further divided into multi-view structure prompts via the same viewpoint cropping as in text prompts (Wang, 2025). The contour maps are extracted as boundaries from the segmentation maps, while the sketches are user-drawn depictions of street scenes. These structure prompts provide important guidance for generating fine structures in the street scenes, such as buildings, roads, *etc.*

## 3.2 PANORAMA–PERSPECTIVE SYNERGY FRAMEWORK

Pretrained latent diffusion models, such as Stable Diffusion (SD) (Rombach et al., 2022), struggle to directly generate consistent panoramic images from multiple viewpoints. Existing iterative or synchronous approaches often fail to maintain loop closure, leading to distorted objects and inconsistent seams. To address this, we design a **two-module framework** consisting of a *panoramic module* and a *multi-view module*, both adapted from the pretrained SD U-Net (see Fig. 2). Unlike PanFusion (Zhang et al., 2024b), which performs bidirectional interaction between the panoramic and multi-view branches and ultimately supervises the panoramic branch with multi-view images, our method **reverses the guidance**: the panoramic branch provides priors to guide multi-view synthesis under structural prompts. In this way, the panoramic module provides *global structural layout*, while the multi-view module focuses on *high-quality perspective synthesis*. Both modules collaborate during the diffusion denoising process, jointly optimizing the multi-view latent representation that is finally decoded into the multi-view outputs.

**Multi-view Module.** Following MVDiffusion (Tang et al., 2023), each generated view has a 90° field of view (FOV) with 45° overlap. A pretrained multi-view SD backbone is employed, and we

integrate structural cues (derived from panoramas) into the U-Net features to reduce distortions and improve inter-view consistency. We further finetune this module on our dataset to better capture domain-specific details before dual-module training.

**Panoramic Module.** This module synthesizes panoramas at $2048 \times 512$ resolution, leveraging LoRA (Hu et al., 2022) for efficient adaptation. To enhance coherence, we apply rotation and cyclic padding during denoising. Unlike PanFusion, our panoramic module does not serve as the final output; instead, it generates structural and layout guidance that enforces consistency across the multi-view generation process. By conditioning the diffusion on panoramic structural information, we improve both the quality of the panoramas and their ability to regularize the multi-view synthesis.

**Structural Information Fusion.** We adopt three methods to integrate structural information into our Panorama–Perspective Synergy Framework. The first method involves encoding the structural information using an encoder to obtain its latent space features, which are then directly added to the latent space noise, denoted as "Add". The second method involves adding viewpoint structural information to the channels during the viewpoint noise input in the multi-view branch, and similarly, adding panoramic structural information to the channels during the panoramic noise input in the panoramic branch, denoted as "Concatenation". The third method involves inputting the viewpoint and panoramic structural information separately into ControlNet (Zhang et al., 2023a). We apply cross-attention between the features and the UNet in the corresponding branch, denoted as "ControlNet+Attn". ControlNet+Attn is the best one because it effectively integrates control mechanisms with attention, enabling more precise feature alignment and improved representation, as shown in the prompt insertion ablation study.

### 3.3 Panorama Alignment Module (PAM)

In our setting, the camera is fixed at a single position and captures different views solely by rotating its orientation. Under this pure-rotation model, the correspondence between perspective views and the panorama is uniquely determined by spherical projection. Based on this prior, we introduce the Panorama Alignment Module (PAM), which explicitly enforces geometric alignment between the panorama and perspective branches, and enables cross-branch feature interaction.

Specifically, given a pixel $(u, v)$ in a perspective view, we first back-project it to normalized camera coordinates as $\tilde{x}_p = K^{-1}[u, v, 1]^\top$, where $K \in \mathbb{R}^{3 \times 3}$ denotes the intrinsic matrix. The direction in world space is then obtained by applying the camera rotation $R \in SO(3)$ and normalization, i.e., $d = \frac{R\tilde{x}_p}{\|R\tilde{x}_p\|}$. The direction vector $d = (d_x, d_y, d_z)^\top$ is further converted to spherical angles $\theta = \arctan 2(d_x, d_z)$ and $\phi = \arcsin(d_y)$, and mapped to panorama coordinates $(u_e, v_e)$ of resolution $(W_e, H_e)$ as $u_e = \frac{\theta + \pi}{2\pi} W_e$ and $v_e = \frac{\pi/2 - \phi}{\pi} H_e$.

This step establishes a **pixel-wise geometric correspondence**: each pixel in the perspective view is uniquely matched to a location in the panorama. The inverse mapping (from panorama to perspective) can be derived analogously by converting $(u_e, v_e)$ to a spherical direction vector and projecting it back with $(K, R)$. In the feature space, PAM avoids interpolation and instead leverages the above mapping as an anchor for cross-branch attention. Concretely, a query token $q$ from the target branch is projected to the corresponding position $(u^*, v^*)$ in the source branch, and cross-attention is performed within its local neighborhood $N(u^*, v^*)$:

$$\text{Attn}(q) = \text{Softmax}\left( \frac{Q_q K_N^\top}{\sqrt{d}} + B_N \right) V_N, \tag{1}$$

where $Q_q \in \mathbb{R}^d$ is the query vector, $K_N, V_N \in \mathbb{R}^{|N| \times d}$ are the sampled key and value features, and $B_N$ is a Gaussian bias emphasizing the projected center.

To adapt PAM to the multi-scale design of the U-Net, we vary the neighborhood size across layers: high-resolution layers ($64 \times 64, 32 \times 32$) adopt small radii ($r = 1, 2$, corresponding to $3 \times 3$ and $5 \times 5$ windows) to preserve local details; the intermediate layer ($16 \times 16$) uses a larger radius ($r = 3$, corresponding to a $7 \times 7$ window) for broader context; and the bottleneck ($8 \times 8$ or $4 \times 4$) directly attends to the entire feature map. This schedule ensures a balance between fine-grained detail preservation and global geometric consistency.

The attention result is further processed by a zero-initialized $1 \times 1$ convolution and fused with the original features in a residual manner:

$$F_{\text{out}} = F_{\text{in}} + \text{Conv}_{1\times1}^{(0)}(\text{Attn}(q)), \tag{2}$$

where $F_{\text{in}}, F_{\text{out}} \in \mathbb{R}^{c \times h \times w}$ denote the input and output features.

By integrating spherical projection with cross-attention, PAM enforces geometry-aware alignment while avoiding interpolation artifacts. When applied at multiple scales of the U-Net, PAM not only preserves the generative capacity of the diffusion backbone, but also significantly improves the consistency between panorama and perspective branches.

## 3.4 Model Training

Unlike the one-stage training methods MVDiffusion (Tang et al., 2023) and PanFusion (Zhang et al., 2024b), our StreetDiffusion adopts a three-stage training strategy, with a particular emphasis on the incorporation of structural information to enhance the quality of the generated results. Our StreetDiffusion borrows the dual-branch architecture from PanFusion and the CAA from MVDiffusion, but differs by introducing panoramic and multi-view structural prompts as conditional inputs to guide the model's spatial layout and geometric consistency during image generation. Additionally, unlike prior work, our model employs a panoramic branch to explicitly assist the multi-view branch, further enhancing the consistency and realism of the generated urban street scenes. The training process is:

- **Stage 1**: We first fine-tune SD on the Street360 dataset to acquire outdoor street scene priors. At this stage, we conduct two types of training on the single-view U-Net: one focuses on single-view street image generation under a normal perspective (FOV=90°, aspect ratio 1:1) to learn local structural priors of street scenes, while the other directly generates panoramic images (aspect ratio 4:1) to learn distribution characteristics under panoramic projection. In this way, the model incorporates both panoramic and view-based structural information, enabling it to generate either single-view or relatively coarse panoramic street images. At this point, although the generated results still suffer from poor consistency, they maintain a structure-conforming appearance within individual views.
- **Stage 2**: We train the multi-view branch to ensure consistency between each view. The multi-view branch learns the relationships between perspectives, improving the consistency of the generated images, though with some trade-offs in realism.
- **Stage 3**: We combine the multi-view and panoramic branches, utilizing the perspective relationships between them and leveraging the global information from the panoramic branch to optimize the global consistency of the multi-view branch. This final stage allows the model to generate panoramic street images that are both consistent and realistic, and most importantly, adhere to the structural constraints. This process fully capitalizes on the significance of structural information, ensuring that the generated images maintain spatial layout, geometric consistency, and coordination between different views.

For the multi-view noises ($\sigma_0 \sim \sigma_7$) and the panoramic noise ($\sigma^*$), we employ the same latent map initialization strategy so that the noise corresponding to spatially aligned positions between the projected multi-view images and the panorama remains consistent. In both Stage I and Stage II, given the panoramic image GT $x^*$ and the multi-view images ($x_0 \sim x_7$) obtained by cropping $x^*$, the losses for the multi-view branch and the panoramic branch are formulated as follows:

$$L^* = \mathbb{E}_{x^*, t, \epsilon^*, y} \left[ \left\| \epsilon^* - \epsilon_\theta^*(z_t^*, t, \tau(y)) \right\|^2 \right], \tag{3}$$

$$L_i = \mathbb{E}_{x_i, t, \epsilon_i, y} \left[ \left\| \epsilon_i - \epsilon_\theta^i(z_t^i, t, \tau(y)) \right\|^2 \right]. \tag{4}$$

In Stage III, we jointly train the panoramic alignment module using both the multi-view branch and the panoramic branch. The corresponding loss is a weighted sum of the two aforementioned losses, formulated as follows: $\mathcal{L} = \mathcal{L}^* + \frac{1}{N} \sum_{i=1}^{N} \mathcal{L}^i$.

# 4 EXPERIMENTS AND RESULTS

## 4.1 DATASET GENERATION.

Currently, there are only a few panoramic and multi-view generation datasets specifically designed for indoor tasks, such as ScanNet (Dai et al., 2017) and Matterport3D (Chang et al., 2017). However, outdoor datasets, especially street scene datasets, remain largely unavailable. (Gardner et al., 2017) contains only 2,100 indoor scenes, while (Zhang and Lalonde, 2017) offered approximately 200 outdoor HDR panoramas. HDR360-UHD (Chen et al., 2022) comprises 1893 outdoor images and 2501 indoor images, while SUN360 (Xiao et al., 2012), similar to HDR360-UHD, is also primarily composed of indoor scenes with relatively few outdoor images. To develop a high-quality street scene generation model, we collected a dataset of 10,000 panoramic/multi-view images covering dozens of regions from the web and combined it with the aforementioned datasets. This resulted in a new high-quality HDR panoramic/multi-view dataset, Street360, containing 10,000 HDR panoramas with resolutions ranging from $4096 \times 2048$ (4K) to $8192 \times 4096$ (8K). For a more comprehensive dataset comparison, please refer to Table 7.

Specifically, most images in our dataset were collected from panoramic resources available on the web, with some examples obtained from public platforms such as Google Maps. We then performed view splitting and conditional generation on these panoramas. In particular, we set the view-splitting parameters following the strategy of MVDiffusion: each panorama was divided into six skybox images, and from the four non-polar views, we further extracted eight perspective images. Based on these splits, we employed OneFormer (Jain et al., 2023) to generate corresponding panoramic segmentation maps, applied Canny edge detection (Canny, 1986) to obtain contour maps, and used BLIP3 (Xue et al., 2024) to produce textual descriptions as conditional inputs, thereby providing diverse supervision signals for subsequent multimodal training.

## 4.2 EXPERIMENT SETTINGS

**Comparison methods.** We compare our StreetDiffusion with the following comparison methods, including multi-view and panorama generation SOTAs: MVDiffusion (Tang et al., 2023), PanFusion (Zhang et al., 2024b), SD+LoRA (Hu et al., 2022; Rombach et al., 2022), Text2Light (Chen et al., 2022; Rombach et al., 2022). In the experiments, we compare with these methods by adding the same structure prompts as our method.

**Implementation details.** For text-conditioned generation, we adopt the same training and test schedules as MVDiffusion and PanFusion. For generations conditioned on segmentation maps and contour maps, we additionally train a ControlNet using the same prompt fusion strategy to ensure fair comparison for all methods.

**Evaluation metrics.** We evaluate the generated results using both automatic metrics and a user study. For **image quality**, we follow prior work and report FID(Heusel et al., 2017), IS, and CLIP Score. To assess **multi-view consistency**, we adopt overlapping PSNR from MVDiffusion. Finally, we conduct a **user study** where participants compare panoramas generated by different methods. More implementation details of each metric are provided in the Appendix.

## 4.3 STREET GENERATION RESULTS

**Quantitative Results.** Table 1 presents the quantitative evaluation results of using different prompts. Using **segmentation maps** as structure prompts, we significantly outperform SD+LORA, PanFusion, and MVDiffusion on FID. Besides, we are the best on the IS and CS metrics, indicating that our model excels in generating diverse objects. In contrast, MVDiffusion tends to avoid generating unexpected objects, often resulting in a large number of repeated items. This repetition may enhance alignment with textual prompts, leading to a slightly higher CS score than SD+LORA and PanFusion, but still lower than ours. Using **contour maps** as structure prompts, our method also achieves the best results among all methods, indicating that our method could perform robustly under different structure prompts. Considering that segmentation maps provide more structural constraints than contour maps, our method with contour maps decreases slightly than with segmentation maps, but still outperforms others.

Table 1: The result comparison on the Street360 dataset with different prompts, where 'seg' and 'cont' refer to using segmentation maps and contour maps as the structure prompts, respectively. Note that the SD+LORA (seg) and PanFusion (seg) methods directly generate full panoramic images. As a result, the overlapping regions between adjacent views are identical after cropping, making the OP_PSNR metric inapplicable for these methods. **Bold** refers to the best method and underline indicates the second best.

| Method | Prompts | FID ↓ | IS ↑ | CS ↑ | OP_PSNR ↑ |
|---|---|---|---|---|---|
| SD+LORA_seg | Text+Seg | 33.78 | 4.83 | 22.48 | - |
| PanFusion_seg | Text+Seg | 27.19 | 4.92 | 22.10 | - |
| MVDiffusion_seg | Text+Seg | 29.18 | 4.78 | 23.12 | 35.66 |
| StreetDiffusion_seg (Ours) | Text+Seg | **10.96** | **6.37** | **24.69** | **39.56** |
| SD+LORA_cont | Text+Contour | 35.41 | 4.75 | 4.75 | - |
| PanFusion_cont | Text+Contour | 28.30 | 3.78 | 17.18 | - |
| MVDiffusion_cont | Text+Contour | 31.72 | 5.03 | 25.12 | 36.21 |
| StreetDiffusion_cont (Ours) | Text+Contour | **12.09** | 5.92 | **25.21** | **39.79** |

**Qualitative Results.** Figs. 18 and 19 show results with **segmentation maps** as structure prompts, highlighting three aspects. *Stylistic similarity*: both MVDiffusion and our method generate outputs close to the ground truth. *Image plausibility*: our framework captures local details and global context, avoiding distorted lines, unnatural objects, and panoramic inconsistencies. *Multi-view consistency*: by integrating cues across views, our method achieves smooth transitions and coherent semantics. Overall, it produces the most realistic panoramas with minimal structural distortions (see Fig. 18). Fig. 20 presents results with **contour maps**. Although less informative than segmentation maps, contours are cheaper to obtain or can be manually sketched. Our method still delivers competitive performance, demonstrating robustness and applicability in scenarios without rich annotations.

**User Study Results.** We conduct a user study to compare the performance of all methods using segmentation maps as structure prompts. We use the same prompt to generate 45 sets of panoramic images. Participants are asked to select the best image based on three criteria: Style Consistency, Reasonableness, and Multi-view Consistency. A total of 100 valid questionnaires were collected,

Table 2: User study results on three metrics: style consistency, realism, and multi-view consistency.

| Method | Style Consist. | Realism | Multi-view Consist. |
|---|---|---|---|
| MVDiffusion | 16.69% | 19.84% | 18.82% |
| PanFusion | 1.78% | 2.45% | 2.88% |
| Ours | **81.53%** | **77.71%** | **78.3%** |

and Table 2 illustrates that over 77% of the users regard our method as generating the best street scene images according to all three criteria. This also proves the advantage of the proposed StreetDiffsuion model in generating more consistent urban street scenes with more realistic object structures.

### 4.4 ABLATION STUDY

**Model architecture.** We compare the results of the standalone panoramic branch, the standalone multi-view branch, and our full Panorama–Perspective Synergy Framework in Table 3. It is observed that the panoramic branch exhibits a worse FID score, indicating that the authenticity of the generated images is insufficient. This is primarily because Stable Diffusion (SD) performs optimally at lower resolutions (e.g., $512 \times 512$), and its performance naturally declines at higher resolutions. However, since the panoramic branch performs denoising on the same noise, the generated images display good overall consistency, suggesting that the panoramic branch possesses certain global consistency features, though it still lacks structural consistency. In contrast, the multi-view branch achieves a better FID score than the panoramic branch, as the image resolution generated by the multi-view branch aligns with SD's strengths. Nevertheless, the consistency between adjacent views is relatively poor due to the simultaneous denoising of multiple noises, leading to suboptimal adjacent view images. Overall, our method not only achieves optimal performance in terms of generation quality but also attains the best consistency results. This proves our model fully leverages SD's powerful generation capabilities, captures global consistency features in the panoramic branch, and effectively controls structural information, thereby generating realistic images that conform to spatial structures and high consistency.

**Prompt insertion.** We conduct an ablation study on different ways of incorporating structural information (Table 4). A simple approach concatenates structural information with the noise input as

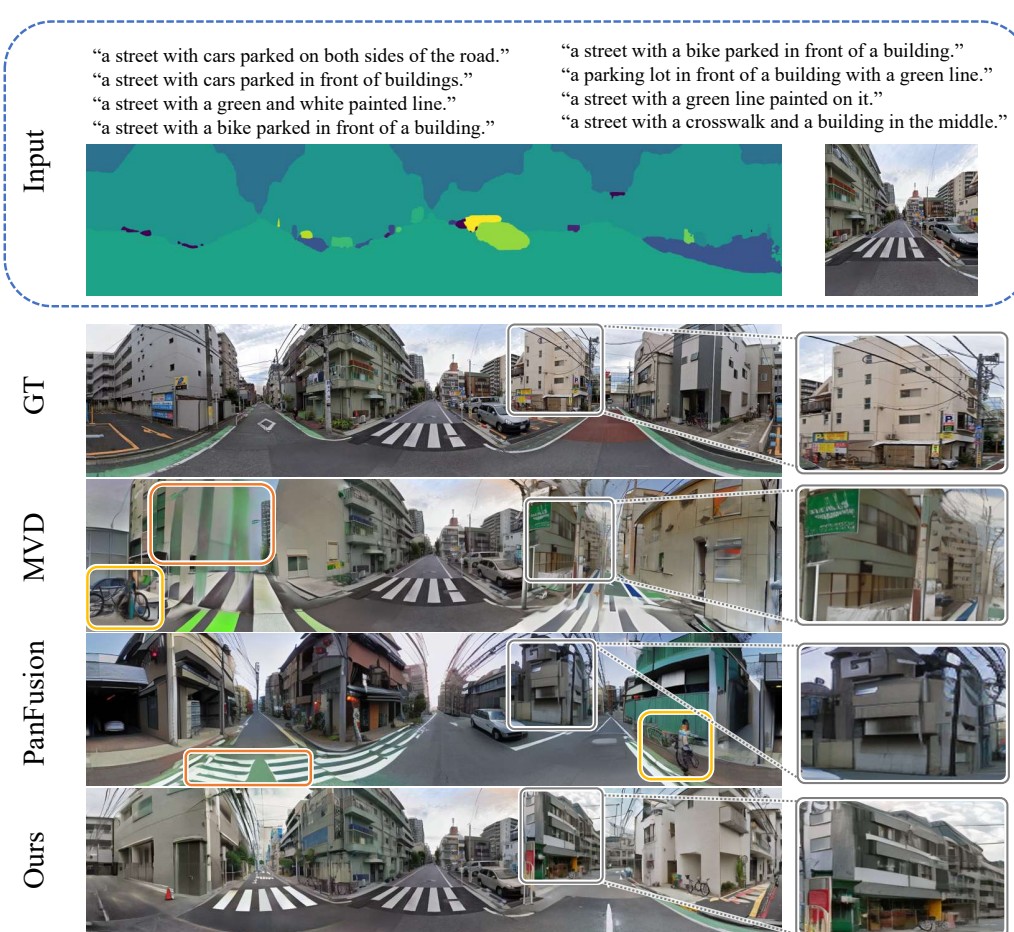

Figure 3: Qualitative comparisons of panorama generation. We present the results obtained by stitching together multi-view images generated by MVDiffusion and StreetDiffusion. Using color-coded boxes, we highlight issues such as distorted lines and unrealistic generated items found in MVDiffusion (Tang et al., 2023) and PanFusion (Zhang et al., 2024b), which are effectively addressed by our method. See more in the Appendix.

Table 3: The model architecture ablation study.

| Method | FID ↓ | IS ↑ | CS ↑ | OP_PSNR ↑ |
|---|---|---|---|---|
| Pano Branch | 35.24 | 4.51 | 20.20 | 30.98 |
| Multiview Branch | 19.26 | 6.21 | 24.58 | 27.29 |
| Both (Ours) | **10.96** | **6.37** | **24.69** | **39.12** |

Table 4: The prompt insertion ablation.

| Method | FID ↓ | IS ↑ | CS ↑ |
|---|---|---|---|
| Add | 23.15 | 5.09 | 20.07 |
| Concatenation | 22.51 | 5.42 | 23.76 |
| Ours (ControlNet+Attn) | **10.96** | **6.37** | **24.69** |

an additional channel. Another approach adopts ControlNet, where each layer output is added to the corresponding layers in the U-Net. Our method instead integrates ControlNet features into the U-Net via cross-attention, which achieves the best results. This demonstrates that cross-attention effectively leverages structural cues, enhancing spatial perception and leading to images that better match the expected outcomes.

**Only using text prompts.** We compare all methods with only text prompts in Table 5. It shows that even without using any structure prompts, our method still achieves the best performance on Street360. This is because the panoramic features in our model effectively guide the generation of multi-view images, leading to better consistency and richer details.

**Street generation with user sketches.** We employ user input sketches as structure prompts for image generation in Fig. 21. Note that the model is trained with contour maps as structure prompts, and the user sketches are only input in the test. It shows that the model is still capable of producing high-

Table 5: The results on the Street360 dataset with only text prompts.

| Method | Prompts | FID ↓ | IS ↑ | CS ↑ | OP_PSNR ↑ |
|---|---|---|---|---|---|
| SD+LORA | Text | 30.19 | 5.06 | 20.46 | - |
| Text2Light | Text | 113.09 | 5.06 | 21.72 | - |
| PanFusion | Text | 18.55 | 4.89 | 21.48 | - |
| MVDiffusion | Text | 24.59 | 5.83 | 24.10 | 38.65 |
| StreetDiffusion (Ours) | Text | **11.95** | **6.02** | **24.93** | **39.12** |

Table 6: The training strategy ablation study. The experiments are conducted on the Street360 dataset with only text prompts.

| Method | FID ↓ | IS ↑ | CS ↑ | OP_PSNR ↑ |
|---|---|---|---|---|
| end-to-end | 22.35 | 4.98 | 22.25 | 38.71 |
| 3-stage (Ours) | **11.95** | **6.02** | **24.93** | **39.12** |

quality results and aligns with the user input well, demonstrating its potential for broad applicability in real-world scenarios, such as assisting in creative creation. **See more results in Appendix.**

**Training Strategy Ablation.** To evaluate the effectiveness of our staged training strategy, we conduct an ablation comparing two training configurations: (i) End-to-End Training, where the entire dual-branch architecture is optimized jointly from scratch, and (ii) Our Three-Stage Training Strategy, which progressively learns panoramic priors, strengthens intra-view consistency in the multi-view branch, and finally aligns both branches jointly. As shown in Table 6 of the supplementary material, the end-to-end variant suffers from significant performance degradation in both visual quality and cross-view consistency. In contrast, our staged strategy achieves noticeably better convergence behavior and higher overall fidelity. These results validate that the progressive learning scheme is not merely an implementation convenience, but a crucial component enabling stable and consistency-aware generation.

## 5 DISCUSSION AND CONCLUSION

In this paper, we propose an urban street scene generation model StreetDiffusion, which can generate highly consistent and realistic multi-view street scene images. The model employs structure prompts and a Panorama–Perspective Synergy Framework to make the Stable Diffusion model deal with the object structure challenges well in street scene generation. We also propose a large multi-view street scene dataset Street360 for studying the issue. The experiments demonstrate that the proposed StreetDiffusion model has a clear advantage on the street scene generation task over existing multi-view generation models designed for indoor or wild applications. Besides, in addition to segmentation maps and contour maps, our model can also accept user-input sketches and generate high-quality images aligning with the user input. The proposed model and dataset shall advance the multi-view generation tasks to more complicated scenarios. In future work, we plan to explore more fine-grained structural guidance, such as user-defined semantic bounding boxes and bird's-eye view (BEV) representations, which may further broaden the applicability of urban street scene generation.

**Ethics Statement.** This work focuses on panoramic street scene generation for research purposes. All datasets used in this study are publicly available and contain no personally identifiable information. We follow ethical practices in dataset usage, respecting licenses and terms of use. Our method is intended for academic research in computer vision and graphics, with no foreseeable direct harm to individuals or communities. We acknowledge potential risks of misuse, such as in creating deepfakes, and emphasize that our contributions are aimed at advancing controllable, structure-guided generation rather than unrestricted content creation. This study complies with the ICLR Code of Ethics.

**Reproducibility Statement.** We provide detailed descriptions of the StreetDiffusion model, training setup, and evaluation protocols in the main text. Additional implementation details, dataset preprocessing steps, and hyperparameter settings are included in the appendix. We will release

anonymized source code and configuration files in the supplementary materials to facilitate reproducibility. Moreover, pseudo-code for the Panorama–Perspective Synergy framework and detailed explanations of structural prompt usage are provided to ensure that all experiments can be replicated.

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

# A APPENDIX

The appendix are structured as follows: Section A.1 provides a detailed description of our experimental setup. Section A.6 presents a comprehensive comparison of our method with other approaches in outdoor scenes, highlighting its advantages. Section A.7 showcases the high-quality results generated by our method.

## A.1 EXPERIMENT DETAILS

As described in Section 4 of the paper, we further elaborate on the experimental setup and baseline methods here.

Table 7: The statistic comparison: Street360 is the first street scene generation dataset with high-resolution images and additional structural information.

| Dataset | Location | Scene type | Image number | Resolution | Condition |
|---|---|---|---|---|---|
| CVRG-Pano | outdoor | countryside road | 600 | 2K | text |
| Matterport3D | indoor | house | 10,800 | 2K | text |
| Scannet | indoor | house | 1,613 | 2K | text |
| HDR360-UHD | indoor, outdoor | house, wild | 4,394 | 4K-8K | text |
| Street360 | outdoor | urban street | 10,000 | 4K-8K | text, seg., contour |

## A.2 DATASET

The Street360 dataset is a large-scale outdoor street-view panoramic dataset containing 10,000 panoramic images from multiple cities. For text-based image generation, we use BLIP-2 Xue et al. (2024) to generate a brief description of the entire image using the prompt. The dataset is split in a 9:1 ratio, with 9,000 images for training and 1,000 for evaluation. Notably, the original Street360 dataset contains blurred areas near the upper and lower edges. To generate more realistic outdoor scenes, we crop out these blurred regions. For text- and structure-based image generation, we use Oneformer Jain et al. (2023) to segment the images, retaining only the segmented images without preserving semantic information. The contour maps are derived by extracting semantic boundaries from the semantic maps. The user-provided sketches were created by three invited participants, who were asked to roughly outline key elements in the panoramic images, such as buildings and vehicles. Compared to contour maps, user sketches are more consistent with real-world drawing styles, enabling our method to be more applicable in practical, real-life scenarios.

A.3   EVALUATION METRICS.

We use image quality metrics and a user study to evaluate the street scene generation results of all methods.

- *Image quality.* Following previous work, FID (Heusel et al., 2017), IS, and CS are used to evaluate the quality of generated images. Fréchet Inception Distance (FID) is widely used to assess image generation quality by measuring the distribution similarity between generated and real images. Inception Score (IS) (Salimans et al., 2016) evaluates the diversity and predictability of generated images. CLIP Score (CS) (Radford et al., 2021) utilizes a pre-trained CLIP model to measure text-image similarity.
- *Multi-View Consistency.* Previous works, such as MVDiffusion and PanoFree, have proposed various metrics for evaluating multi-view consistency. Following MVDiffusion, we adopt overlapping PSNR as a metric to assess the consistency between multi-view images. Specifically, we compute PSNR over the overlapping regions of the ground truth images and further calculate the overlapping PSNR for the generated images in the same regions. However, overlapping PSNR mainly measures pixel-level differences and does not fully capture the structural and textural relationships in the images. To address this, we propose a new metric overlapping SSIM to quantify multi-view images in terms of both structural consistency and texture consistency.
- *User Study.* However, existing metrics do not fully capture all aspects of human perception of quality. Therefore, we set up a voting webpage displaying side-by-side panoramic scenes generated using the same text input—one from our method and the others from baseline methods. We survey anonymous voters and ask them to choose the generated image with higher quality and better scene structure.

A.4   IMPLEMENTATION AND TRAINING DETAILS

We implemented our model in PyTorch and developed it based on Stable Diffusion provided by Diffusers. Following MVDiffusion Tang et al. (2023), we trained the model for 10 epochs using the AdamW optimizer with a batch size of 4 and a learning rate of 2e-4, and we adopted the DDIM sampler Stan et al. (2023) with 50 steps during inference. Additionally, when training the extra ControlNet Zhang et al. (2023a) for text-structure conditioned generation, all training was conducted on four A6000 GPUs, with the total training time for both text and structure conditions amounting to approximately 23 hours.

In Section 3.4, we introduced the three-stage training process of the model. Below, we provide a more detailed description of the specific training configurations for each stage.

**Training Stage One.** We mainly train on single-view images, performing image generation under conditions using only text prompts as well as using both text and structural prompts (such as semantic maps, contour maps, or user sketches). The training image resolution is $512 \times 512$, with 10 epochs of training. The optimizer used is AdamW, with a batch size of 4 and a learning rate of 1e-5. Training is conducted on four NVIDIA A6000 GPUs.

**Training Stage Two.** Based on the SD model trained in Stage One, which already has outdoor scene priors, we continue to train the multi-view branch and the panoramic branch. For the multi-view branch, eight viewpoint images (with a field of view (FOV) of 90° and an overlap angle of 45° between adjacent images) are input into the multi-view diffusion model simultaneously to learn the geometric consistency across views. In this stage, we follow the MVDiffusion training setup: the training image resolution is $512 \times 512$, trained for 10 epochs, with AdamW optimizer, batch size 4, and learning rate 2e-4. For the panoramic branch, we use panoramic images and their corresponding structural prompts (including panoramic contour maps and panoramic text descriptions) as input, aiming to fine-tune the SD model to generate high-quality panoramic images. Here, training is performed on multiple scales with an aspect ratio of 4:1, at resolutions such as ($2048 \times 512$, $1024 \times 256$, down to $256 \times 128$), trained for 10 epochs using AdamW optimizer with a learning rate of 1e-5.

**Training Stage Three.** We integrate the panoramic branch and the multi-view branch by freezing the Unet parameters in both branches and training only the interaction module between them. The core objective of this stage is to enable the model to effectively leverage the global contextual in-

Table 8: Results on the indoor dataset Matterport3D (Chang et al., 2017) multi-view generation dataset with only text prompts.

| Method | FID ↓ | IS ↑ | CS ↑ |
|---|---|---|---|
| Text2Light (Chen et al., 2022) | 43.66 | 4.92 | 25.88 |
| SD+Lora (Rombach et al., 2022; Hu et al., 2022) | 23.02 | 6.58 | 28.6 |
| PanFusion (Zhang et al., 2024b) | 19.88 | 6.50 | 24.98 |
| MVDiffusion | 21.44 | 7.32 | **30.04** |
| StreetDiffusion (Zero-shot) | 16.26 | 6.31 | 24.51 |
| StreetDiffusion (Fine-tuned) | **13.16** | **7.38** | 29.38 |

formation provided by the panoramic branch to enhance the generation quality of the multi-view branch, thereby further improving spatial consistency and realism of the generated images. In this stage, the inputs are panoramic contour maps and panoramic text descriptions, which are segmented into multi-view inputs using a tool, enabling multi-view image generation guided by panoramic conditions.

## A.5 IMPLEMENTATION DETAILS OF COMPARISONS

**MVDiffusion** Tang et al. (2023): we followed the original settings and further fine-tuned its pre-trained model on the Street360 dataset to enhance its performance in multi-view image generation. After fine-tuning, we tested the model's generated multi-view images and then stitched them together to form complete panoramic images.

**SD+LoRA** (Pano branch) Dhariwal and Nichol (2021); Hu et al. (2022): We fine-tuned Stable Diffusion using 9,000 panoramic images from the training dataset, each with a resolution of 512 × 2048. During fine-tuning, only the UNet layer of Stable Diffusion was trained, while the VAE layer remained frozen. The training was conducted using the AdamW optimizer with a learning rate of 1e-6, a batch size of 4, and was performed in parallel on four A6000 GPUs.

**PanFusion** Zhang et al. (2024b): To ensure text consistency across different views, we concatenated the text descriptions of each perspective image into a complete text input, effectively guiding the model to generate coherent panoramic images. Similar to the MVDiffusion method, we also fine-tuned the model on the Street360 dataset to further improve the quality and consistency of the generated panoramic images. After fine-tuning, we tested the generated panoramas and then split them into multiple perspective images to ensure that the final results met our expectations.

The above training process applies only to the case of text-only generation. For scenarios involving structural prompt conditioning, we employ a unified ControlNet trained on the corresponding conditions for all methods.

## A.6 EXTRA EXPERIMENTS ON INDOOR SCENE GENERATION

We perform the proposed method on the indoor scene multi-view generation task, and compare with other methods in Table 8. For a fair assessment, we report two versions of our model: (i) **StreetDiffusion (Zero-shot)**, evaluated directly on indoor scenes without any fine-tuning, and (ii) **StreetDiffusion (Fine-tuned)**, trained on Matterport3D for improved indoor adaptation. **StreetDiffusion (Fine-tuned)** achieves the best FID metric, and relatively worse IS and CS metrics than the indoor scene generation SOTA method MVDiffusion. This still proves the effectiveness of the proposed model for different multi-view scene generation tasks.

## A.7 MORE VISUALIZATION RESULTS

Here are some visualization results comparing our method with the current state-of-the-arts (SOTAs) on our Street360 dataset. Please see Figures 4-22 for more details. Overall, our proposed method can achieve more realistic street scene generation results with various structure prompts: segmentation maps, contour maps, and user sketches.

## A.8 ADDITIONAL ANALYSIS ON VIEW–TEXT MISMATCH.

Most of our generated multi-view results match well with the corresponding local text prompts. However, a few cases still show incomplete alignment. Here we briefly explain why such situations occur.

Our eight generated views have 45° overlaps, so adjacent perspectives inevitably contain similar scene content. The BLIP captions extracted from these overlapping views also tend to focus on the same main objects. As a result, the text descriptions of neighboring views often mention overlapping elements. During generation, the model needs to maintain structural consistency across all views. To avoid duplicating objects across multiple perspectives—which would lead to unrealistic geometry—the model sometimes makes conservative choices when integrating information from similar prompts.

A concrete example can be found in Fig. 22. The fourth generated view corresponds to the prompt "a bedroom with a bed and a chair in it." Although the prompt mentions a bed, the bed does not appear in the fourth view. Instead, it is clearly visible in the right-adjacent view (the fifth image). In fact, the BLIP captions for the 4th, 5th, and 6th views all mention the bed. It would be physically inconsistent if a single bed appeared simultaneously in all three views. To preserve a coherent spatial layout, the model generates the bed only in the appropriate view(s), even if one of the prompts suggests otherwise.

In short, these mismatches arise from the balance between following local text descriptions and maintaining globally coherent multi-view geometry. They are a side effect of enforcing structural consistency rather than a failure to interpret the prompts.

"a street with cars parked in front of a building. "  "a street with a building in the background."

"a street with cars parked on both sides of the road. "  "a man is walking down the street in front of a building. "

"a white van is parked in front of a building. "  "a man is walking down the street. "

"a building with a white wall and a sign on the side. "  "a building with cars parked in front of it. "

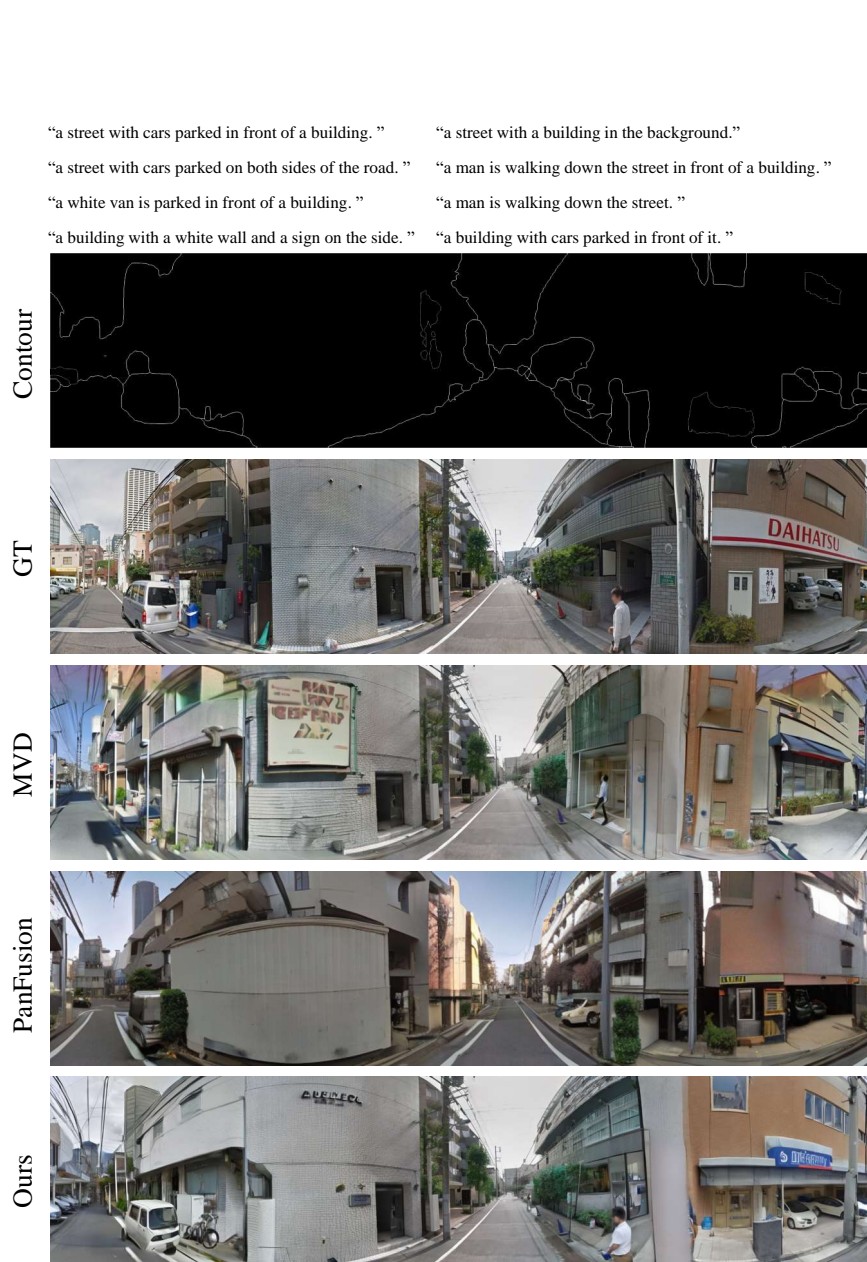

Figure 4: Example of using contour maps as structure prompts.

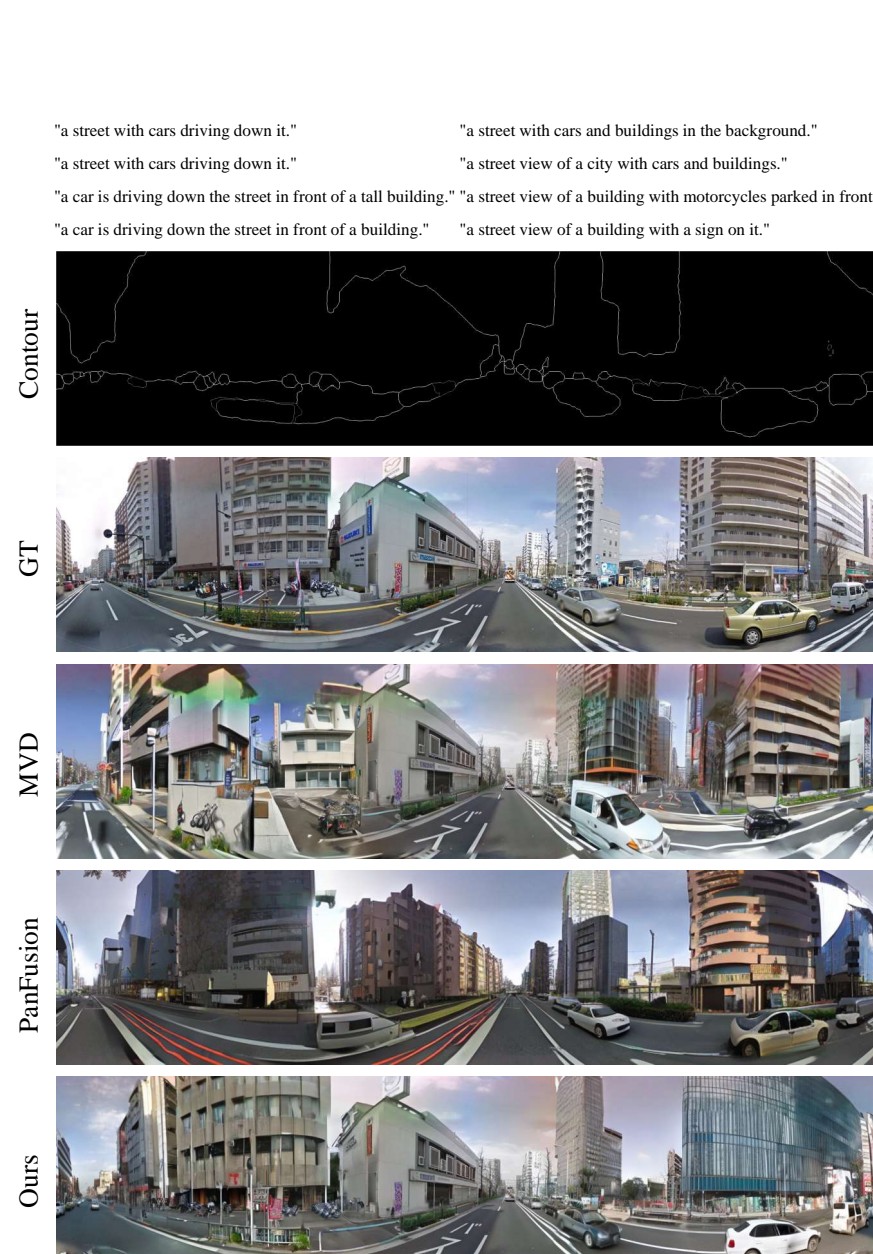

Figure 5: Example of using contour maps as structure prompts.

"a google street view image of a building with a large glass window"   "a street with people walking on it and a building in the background"
"a truck is parked on the side of a street"   "a street view of a building with a sign that says taiwan"
"a large building with a sign that says japanese restaurant"   "a street with a building in the background and a car parked in front of it"
"a street with many buildings and people walking down the street"   "a large building with a large glass door"

Figure 6: Street scene generation results with only text conditions.

"a street with a building in the background"       "a street with a building on one side and a car on the other"
"a narrow alley with a vending machine and a building"       "a street with cars parked in the middle of it"
"a building with a large awning and a sign on the side"       "a car parked in front of a building with a blue balcony"
"a street view of a building with a sign on the side"       "a building with a fence and a tree in the middle"

Figure 7: Street scene generation results with only text conditions.

"a street with a person walking down the middle of it"   "a street with a building in the background and cars parked on the side"
"a street with tall buildings and a person walking down the middle"   "a street with a car parked in front of a building"
"a street view of a building with a tree in the middle"   "a google street view image of an apartment building"
"a building with a blue roof and a sign on the side"   "a google street view image of a residential area"

Figure 8: Street scene generation results with only text conditions.

Figure 9: Example of using user input sketch as structure prompts.

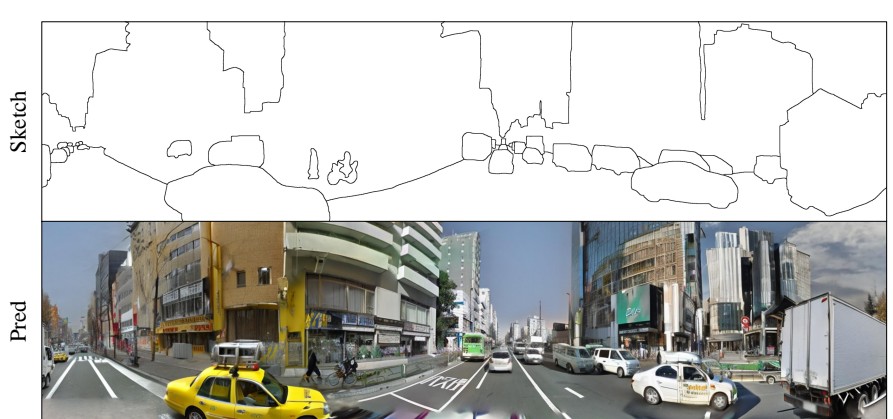

Figure 10: Example of using user input sketch as structure prompts.

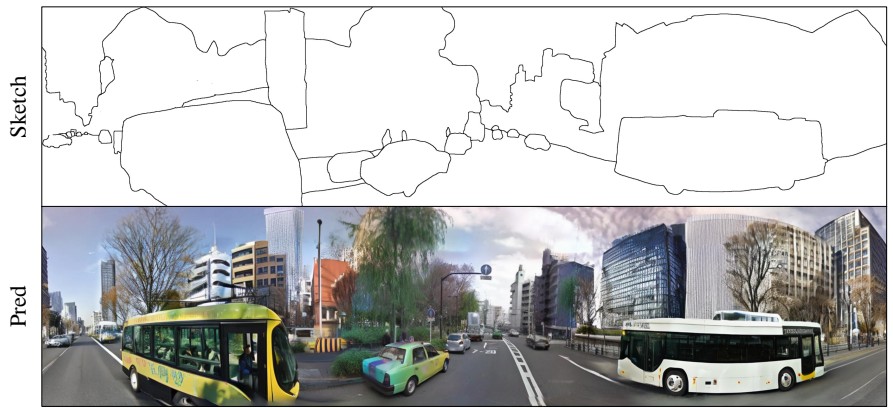

Figure 11: Example of using user input sketch as structure prompts.

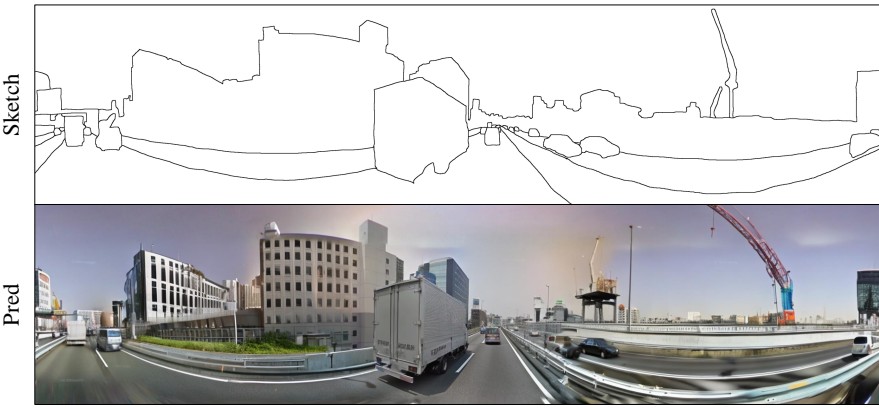

Figure 12: Example of using user input sketch as structure prompts.

"a truck driving down a highway with tall buildings behind."   "a street view image of a highway with cars and traffic."

"a street view image of a highway with cars and trucks."   "a street view of a highway with cars and buildings."

"a car driving on a bridge with a crane in the background."   "a car driving on a highway near a large building."

"a car driving on a highway with a crane in the background."   "a truck is driving down the road near a tall building."

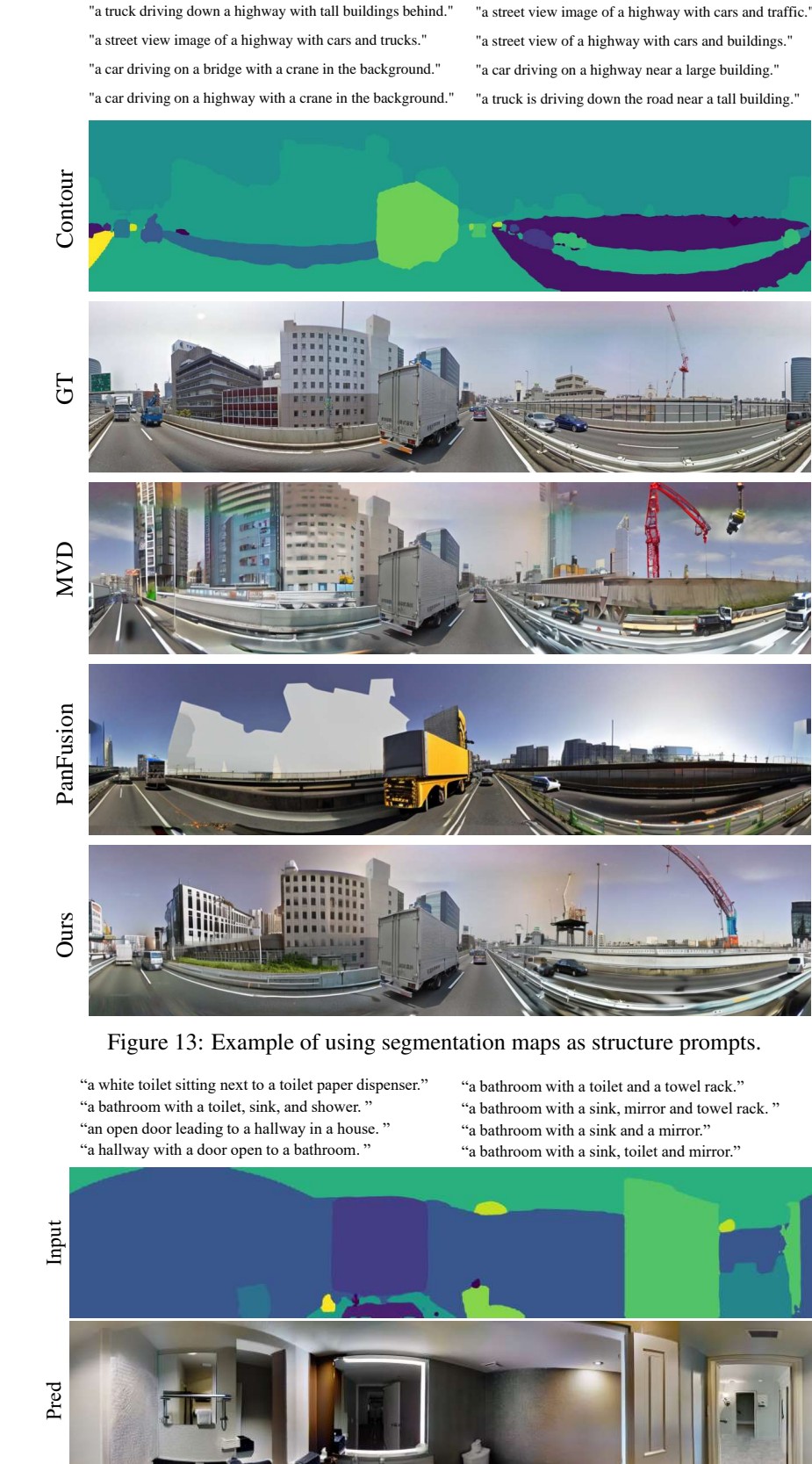

Figure 13: Example of using segmentation maps as structure prompts.

"a white toilet sitting next to a toilet paper dispenser."   "a bathroom with a toilet and a towel rack."
"a bathroom with a toilet, sink, and shower. "   "a bathroom with a sink, mirror and towel rack. "
"an open door leading to a hallway in a house. "   "a bathroom with a sink and a mirror."
"a hallway with a door open to a bathroom. "   "a bathroom with a sink, toilet and mirror."

Figure 14: Indoor generation results using segmentation maps as structural prompts.

"a dining room with a large painting on the wall."
"a view of a dining room from the kitchen. "
"a view of a kitchen from the hallway. "
"a kitchen with a white counter top and shelves."

"a kitchen with a sink and a counter top."
"a kitchen with a sink and a mirror. "
"a kitchen with white counter tops and cabinets."
"a view of a living room through a mirror."

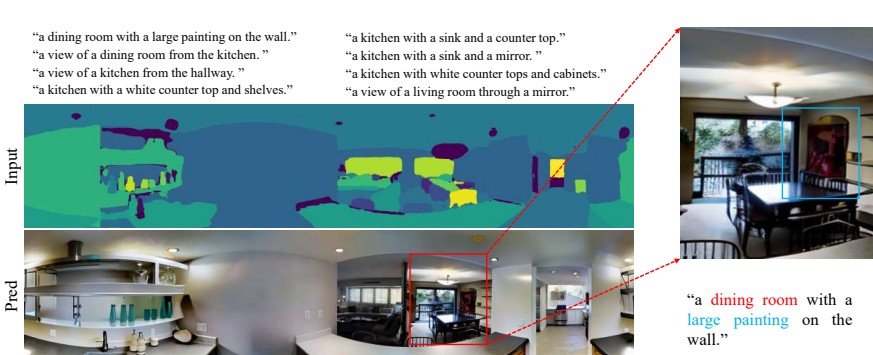

"a dining room with a large painting on the wall."

Figure 15: Indoor generation results using segmentation maps as structural prompts.

"a picture of a living room with a painting on the wall."
"a bathroom with a toilet and a shower. "
"a view of a living room through a sliding glass door. "
"a bedroom with a bed and a chair in it."

"a bedroom with a bed and a desk in it."
"a bedroom with a bed and a couch in it. "
"a living room with a couch and a television."
"a living room filled with furniture and a painting."

Figure 16: Indoor generation results using segmentation maps as structural prompts.

"a dining room with a glass table and white chairs."
"a living room with a couch and a painting on the wall. "
"a living room with a black floor and white walls. "
"a bathroom with a sink and a mirror."

"a kitchen with a white counter and black stools."
"a kitchen and living room with a long counter. "
"a living room with a television and a dining table."
"a living room with a glass table and white chairs."

Figure 17: Indoor generation results using segmentation maps as structural prompts.

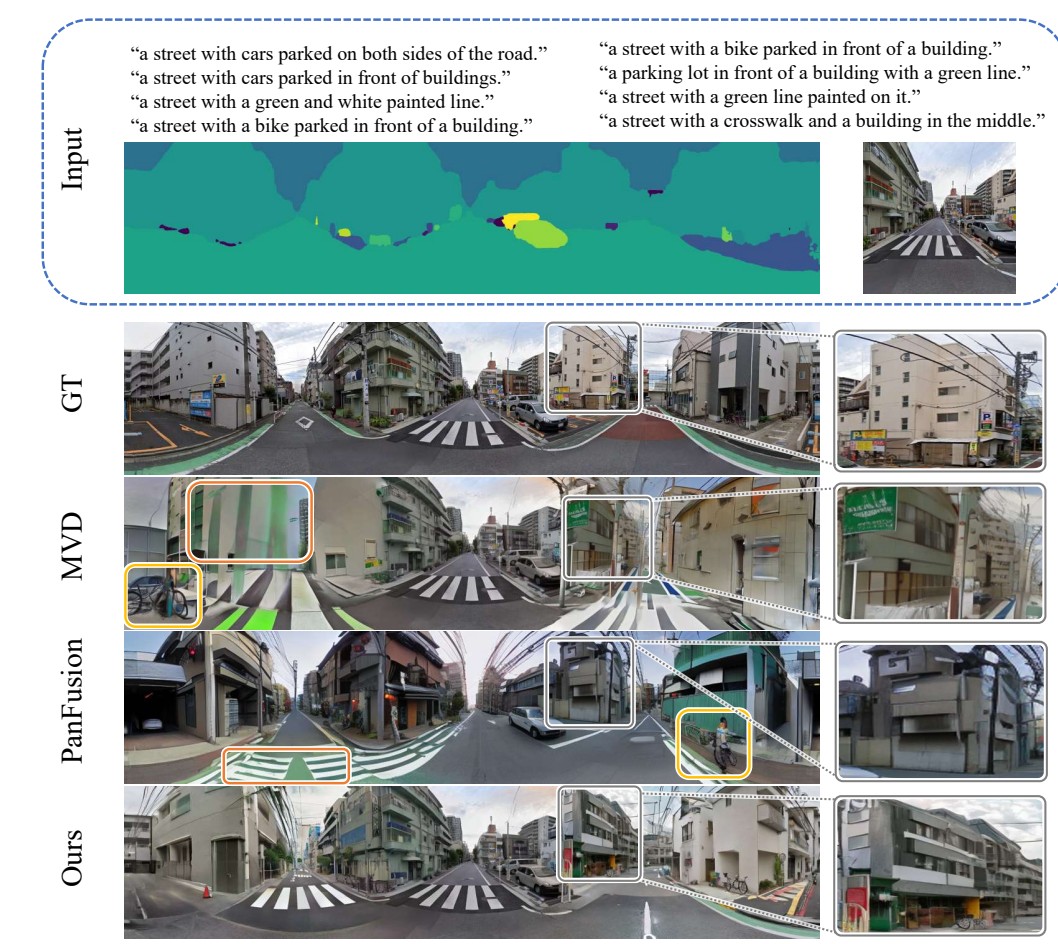

Figure 18: Qualitative comparisons of panorama generation. We present the results obtained by stitching together multi-view images generated by MVDiffusion and StreetDiffusion. Using color-coded boxes, we highlight issues such as distorted lines and unrealistic generated items found in MVDiffusion Tang et al. (2023) and PanFusion Zhang et al. (2024b), which are effectively addressed by our method.

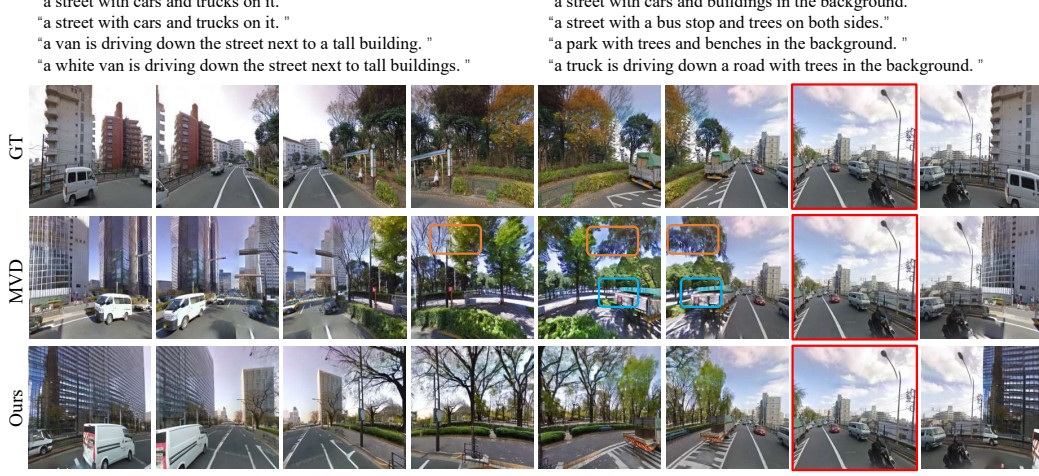

Figure 19: The multi-view street generation results, with the input view highlighted in red boxes. The comparison produces implausible results: trees seemingly growing in the sky (see yellow boxes), and an incomplete truck (see blue boxes). See panorama generation results in other figures.

"a street with cars and trees on both sides."
"a street with cars and trees in the background. "
"a street with cars parked on it and trees in the background."
"a street with cars parked in front of buildings. "

"a street with cars parked on both sides of the road."
"a green taxi is parked on the side of the road. "
"a green car parked next to a fence."
"a street with a fence and trees in the background."

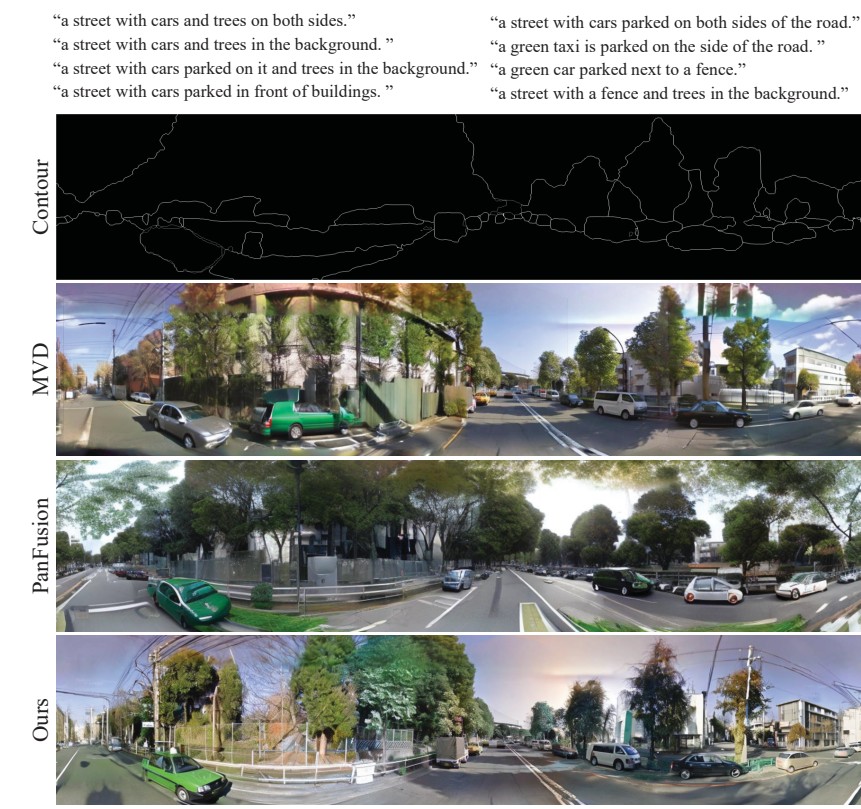

Figure 20: Street scene generation with contour. Our generated street scenes are more realistic and align with the input contour map better.

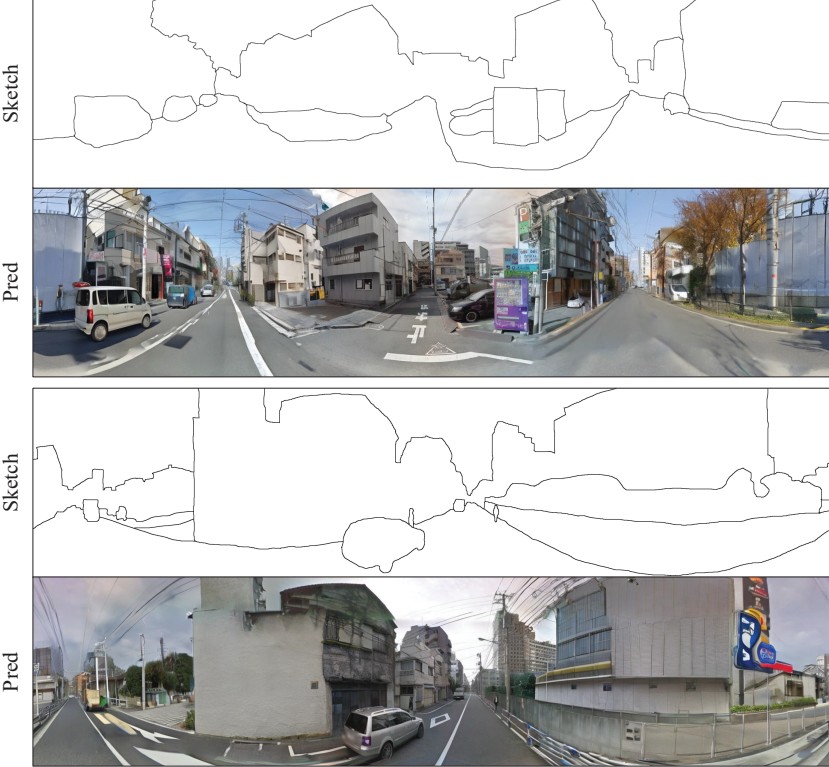

Figure 21: The result of street scene generation with user input sketches. The generated street scenes generally align with user sketches.

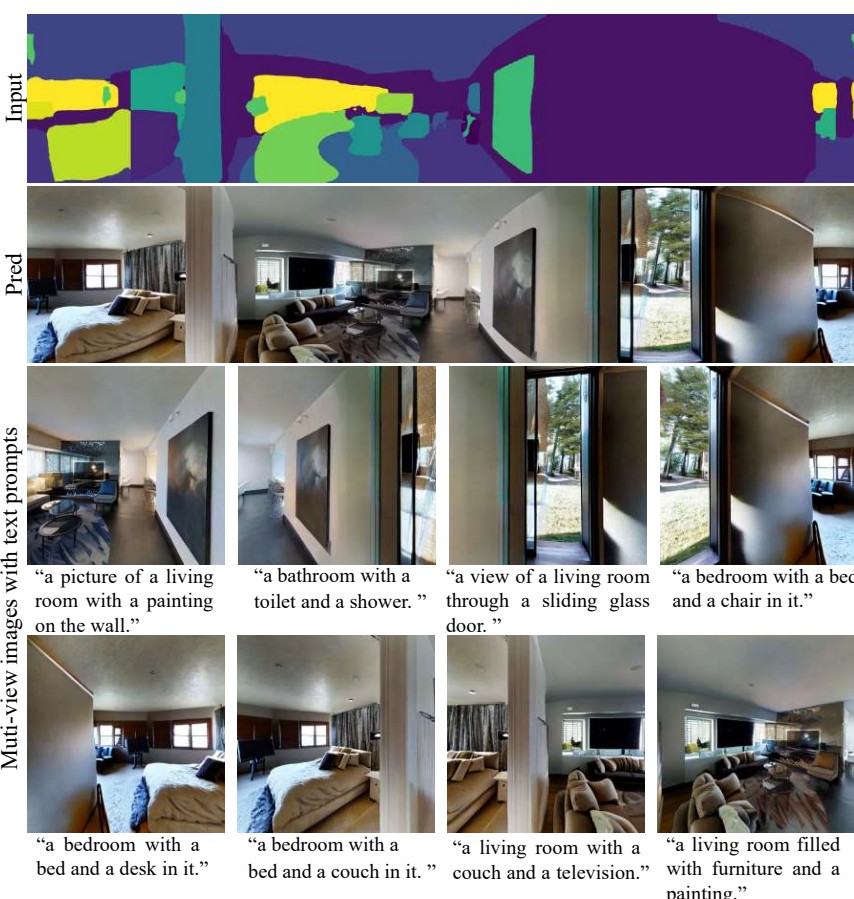

Figure 22: This visualization provides a more detailed illustration of the results shown in Fig. 16. Note that the multi-view images are generated outputs, whereas the text shown here is the input condition rather than a description of the generated results.

