# OpenReview forum: "StreetDiffusion: Street Scenes Generation via Multi-view Stable Diffusion with Structure Prompts"
_ICLR.cc/2026/Conference — Submitted to ICLR 2026_

### Official Review · Reviewer_Ynee · 2025-10-21

**Soundness:** 1
**Presentation:** 2
**Contribution:** 1
**Rating:** 2
**Confidence:** 4

**Summary:**

The paper presents StreetDiffusion, a diffusion-based framework for generating street scenes conditioned on various structural prompts. The authors introduce a PAM to ensure multiview and global consistency. In addition, they propose a new dataset, Street360, which includes panoramic street images and multimodal annotations. However, the proposed task appears to have limited significance, and the overall performance of the method is underwhelming.

**Strengths:**

1. The proposed Street360 dataset could potentially enrich data diversity for street scene generation and related multimodal research.
2. The proposed PAM module is a reasonably good design, integrating global (panoramic) and local (multi-view) cues through attention mechanisms to maintain consistency.

**Weaknesses:**

**1. Poor qualitative results**
* In Figure 1, there is no visible geometric or structural correspondence between the structure prompts(left) and the generated results(right).
* In Figure 3, the given prompts (e.g., “a street with a bike parked in front of a building”, “a street with a green line painted on it”) are not reflected in the generated images. The adherence to prompts is significantly weaker compared to baselines such as MVD and PanFusion.

**2. The task itself seems ill-motivated**
* The proposed problem of generating images from texture or structure prompts can already be effectively addressed by existing methods such as ControlNet.
* The authors seem to have transformed an otherwise straightforward texture-conditioned generation task into a more complex panoramic stitching problem, yet the motivation behind this design choice is not clearly articulated.
* To justify the necessity of this new task, the authors should demonstrate that a baseline like ControlNet (conditioned on panoramic structure maps and text) fails to perform well, thereby motivating the proposed formulation.

**3. Overstated contribution in dataset design**

The claimed contribution in constructing a “street scene” dataset is overstated, as existing datasets such as KITTI and VIGOR already include rich street-level scenes that can serve similar purposes.

**4. Limited performance and generalization**
* The proposed method does not achieve competitive results on the Matterport3D benchmark, which raises concerns about generalization.
* The authors are encouraged to include qualitative results on Matterport3D to demonstrate the general applicability and robustness of their approach.

**5. Ambiguous design choice**

Why are panoramic images divided into eight perspective views rather than the conventional six cube-map faces? The motivation and advantage of this specific decomposition should be clarified.

**Questions:**

The problem corresponds one-to-one with the weakness:
1. How do the authors explain the poor structural and semantic alignment between prompts and generated results? Why does the model fail to capture explicit objects described in the prompt?
2. What is the motivation for defining this task when similar outcomes can be achieved using ControlNet? Can the authors show that existing diffusion-based methods cannot solve this problem effectively?
3. How does Street360 differ fundamentally from existing datasets like KITTI or VIGOR, beyond including panoramic data?
4. Why does the method not outperform existing approaches on Matterport3D? Could the authors provide visualizations on Matterport3D to verify generalization?
5. What is the reasoning behind splitting panoramic images into eight perspective views instead of six? Does this improve consistency or model efficiency?

**Details Of Ethics Concerns:**

Non-existent

---

> ### Author Response · Authors · 2025-11-26
> **Response to Reviewer Ynee**
>
> ### **Q1: Figure presentation issues in Figures 1 and 3.**
>
> **A1:**  For Figure 1, our original intention was to show an example of our dataset samples (left) and generated results (right), without a one-to-one correspondence between them.
>
> For Figure 3, the issue was caused by using an incorrect sample in the submission version. We have updated the figure in the revised manuscript, which now correctly reflects the geometric and textual consistency between the text prompts and the generated results.
>
> ### **Q2: The task itself seems ill-motivated: Add a baseline like ControlNet (conditioned on panoramic structure maps and text).**
>
> **A2:**  We have already conducted experiments by combining existing methods with ControlNet using the same prompts as ours (see our Implementation details in Lines 357-359), as shown in Table 1 (Column 2) and the Appendix figures. The results clearly show that simply integrating ControlNet does not yield satisfactory panoramic results — the generated images suffer from inconsistent geometry and poor structural fidelity.
>
> Moreover, generating a high-resolution panorama directly from Stable Diffusion is inherently difficult, since the pretrained model is limited to 512×512 resolution. Even with LoRA fine-tuning, the generated panoramas exhibit blurred local textures and inconsistent details. Therefore, we reformulate the task as a multi-view generation problem, where each sub-view maintains high local fidelity, and cross-view consistency can be enforced through our dual-branch design. This formulation effectively addresses the limitations of direct panoramic generation and provides superior results compared to existing methods that incorporate ControlNet for structural conditioning.
>
> ### **Q3: Overstated contribution in dataset design.**
>
> **A3:**  While it is true that existing datasets such as KITTI and VIGOR provide street-level scenes, our dataset differs in several key aspects. First, KITTI only captures front-view images with a limited horizontal FoV and is not designed for a full 360° panorama or multi-view synthesis. VIGOR provides 360° panoramas, but its primary purpose is cross-view geo-localization rather than generating multi-view consistent images. In contrast, our dataset is specifically constructed for multi-view and panoramic street scene generation, with paired multi-view images and high-quality structural annotations (semantic, contour maps, and text), which are crucial for training models to achieve consistent and realistic multi-view synthesis.
>
> ### **Q4: The method does not achieve competitive results on the Matterport3D.**
>
> **A4:**  Our method is primarily designed for complex outdoor street scenes, where it leverages street-specific priors in both multi-view and panoramic branches. Indoor environments have been well studied in prior work, and our framework was not fine-tuned on Matterport3D, so the reported results reflect a zero-shot evaluation rather than the intended use case. In Table 8 , we provide updated metrics based on indoor-trained models for a fairer comparison. Among them, the FID, IS, and CS metrics of our method have been updated to 13.16, 7.38, and 29.38. Additionally, in the appendix, we include qualitative results of our method generating indoor panoramic images.
>
> | Method                       | FID ↓     | IS ↑     | CS ↑      |
> | ---------------------------- | --------- | -------- | --------- |
> | Text2Light                   | 43.66     | 4.92     | 25.88     |
> | SD+Lora                      | 23.02     | 6.58     | 28.6      |
> | PanFusion                    | 19.88     | 6.50     | 24.98     |
> | MVDiffusion                  | 21.44     | 7.32     | **30.04** |
> | StreetDiffusion (Zero-shot)  | 16.26     | 6.31     | 24.51     |
> | StreetDiffusion (Fine-tuned) | **13.16** | **7.38** | 29.38     |
>
> Importantly, our core contribution remains in outdoor multi-view consistent generation, where the model achieves state-of-the-art performance.

---

> > ### Author Response · Authors · 2025-11-26
> > **Response to Reviewer Ynee (2)**
> >
> > ### **Q5: Why are panoramic images divided into eight perspective views?**
> >
> > **A5:**  (1) Regarding the resolution issue, our current model already generates 2K panoramic images(Panfusion 1k), which is higher than most existing methods. While the output resolution can indeed be further enhanced by super-resolution, our approach directly synthesizes high-quality details during generation. The current resolution is mainly limited by GPU memory (48 GB). With a larger memory capacity, our framework can scale up to 1024×1024 per view, enabling 4K–8K panoramas accordingly.
> >
> > (2)) Our eight-view design corresponds to the four horizontal cube faces but is further split into eight overlapping perspective views. Unlike standard cube-map generation (e.g., CubeDiff), where faces only meet at boundaries, the overlapping regions in our views provide strong multi-view consistency constraints, which significantly reduce cross-view discontinuities. In addition, since our dataset is captured by vehicle-mounted cameras, the top and bottom poles mostly contain non-informative or irregular content (e.g., car roof, vehicle hood). Therefore, we remove these poles and focus the generation on the meaningful horizontal region.

---

> > > ### Comment · Reviewer_Ynee · 2025-11-26
> > >
> > > Thanks the authors for the detailed clarifications.
> > >
> > > Regarding Q1, the title in Figure 1 reads: “The left side of the figure illustrates representative sample and structure prompts from our proposed Street360 dataset, while the right side presents representative generated results.” This description is highly ambiguous, and the authors should clarify that the left and right sides are not correspondingly paired.
> > >
> > > Still in Q1, in Figure 3, the updated visualizations indeed show some improvement. However, I notice that inconsistencies between the text and the generated images remain common. For example, in Figure 15, the prompt “a dining room with a large painting on the wall” does not appear to match the generated output. The authors need to analyze the underlying causes of such mismatches. One potentially useful approach is to visualize the pinhole images corresponding to each local prompt in order to identify where the discrepancy arises.
> > >
> > > Regarding Q5, the authors claim that “the overlapping regions in our views provide strong multi-view consistency constraints.” However, in the bottom row of visualizations in Figure 19, only the 4th–6th images from left to right appear to contain such overlapping regions. Similar overlapping areas seem uncommon in the other images.

---

> ### Author Response · Authors · 2025-11-27
> **Response to Reviewer Ynee**
>
> **Thank you helping us improve our paper. We have revised the manuscript accordingly.**
>
> ### Q1 — Clarification of Figure 1 & text–image mismatch in Figure 15
>
> 1. **Figure 1 caption clarity.**
>    We agree that the original caption was ambiguous. In the revised paper, we have updated the caption to explicitly state that **the left and right panels are not paired examples** (Lines 28-31). The left panel shows representative samples and structure prompts from our Street360 dataset, while the right panel shows representative generated results.
>
> 2. **Figure 15 text–image mismatch.**
>    After re-checking the examples, we found that the content corresponding to the caption *“a dining room with a large painting on the wall”* **is indeed present** in the generated output, although it may appear subtle in the global panorama. For Figure 15, we have updated the visualization and provided an enlarged view of the region corresponding to the prompt *“a dining room with a large painting on the wall”*. We additionally added Figure 22, which visualizes the pinhole-perspective views corresponding to each local prompt.
>
> 3. **Additional analysis.**
>    In the revised appendix, we include a new subsection titled **“Additional Analysis on View–Text Mismatch”**(Lines 864-885), where we examine this issue in more detail, discuss why such mismatches may occur under multi-view structural constraints, and provide concrete examples to illustrate these cases. Most views are consistent with their prompts, but mismatches can arise due to the strong structural consistency enforced across overlapping viewpoints. Because adjacent views share roughly 45° of overlap, BLIP often produces similar object descriptions for neighboring views. To avoid placing the same large object repeatedly across multiple adjacent views—which would lead to unrealistic scenes—the model may generate the object only in the most structurally suitable view. For example, in Figure 22, although the prompt for View 4 mentions “a bedroom with a bed and a chair,” the bed appears primarily in View 5 (and partially in Views 5–6) to maintain a coherent spatial layout. This explains the few cases where text and view are not perfectly aligned.
>
> ---
>
> ### Q5 — Overlapping regions in Figure 19
>
> We appreciate your careful observation. You are correct that the previous version mistakenly included **misordered images**, which obscured the overlapping regions we referred to.
> We have **replaced Figure 19** with the corrected version in the revised manuscript, where the overlapping views (especially those in the 4th–6th columns) are clearly visible and consistent with our explanation.

---

> > ### Author Response · Authors · 2025-11-29
> > **Response to Reviewer Ynee**
> >
> > We thank the reviewer for the detailed and careful examination of our manuscript. We have addressed all raised concerns in the revision. Specifically, we clarified the intended meaning of Figure 1, updated the visualization and explanations for the text–image mismatch observed in Figure 15, and provided enlarged views together with a new multi-view illustration (Figure 22). We also added an “Additional Analysis on View–Text Mismatch” section in the appendix to more thoroughly discuss how multi-view structural constraints may lead to occasional prompt inconsistencies. Finally, we corrected the misordered images in Figure 19 to clearly show the overlapping regions referenced in the text.
> >
> > We hope these clarifications and updates help improve your assessment. If you have further questions or would like additional analyses, we would be glad to continue refining the paper.

---

### Official Review · Reviewer_dsEf · 2025-10-31

**Soundness:** 3
**Presentation:** 3
**Contribution:** 2
**Rating:** 4
**Confidence:** 3

**Summary:**

This paper tackles the problem of generating high-quality, multi-view consistent urban street scenes, a domain where existing text-to-multi-view methods struggle with complex layouts and dynamic elements. The authors identify two key challenges: a lack of complex urban datasets and the tendency of current models to produce structural artifacts.

To address this, they make two core contributions:

1. Street360 Dataset: A large-scale dataset of over 10K real-world street panoramas, complete with text prompts and structural annotations (e.g., segmentation maps).

2. StreetDiffusion Model: A novel framework that synergizes a panoramic branch and a perspective branch. Its key innovation is the Panorama Alignment Module (PAM), which uses structural prompts and geometry-aware attention to enforce global consistency and local detail.

Experiments show the proposed method outperforms existing models in both quality and multi-view consistency for street scene generation.

**Strengths:**

+ A new framework: The "Panorama–Perspective Synergy Framework," which integrates a global panoramic branch and a local perspective branch, guided by structural prompts (e.g., segmentation maps, sketches), to enhance consistency and realism.

+ A new dataset: Street 360, a large-scale dataset of 10K multi-view and panorama images to support research in this specific domain.

+ Superior performance: Experimental results show that StreetDiffusion outperforms existing multi-view diffusion models on the task of street scene generation.

**Weaknesses:**

1. The author argues that previous works can only generate images with limited resolution (1024x512) due to their architecture design, while this paper is able to generate high-resolution images (2048x512).
However, I don't think that this is a resolution of previous works. The resolution can be easily handled by a downstream super-resolution network. Furthermore, a typical panorama image holds a 180-degree vertical FoV and 360-degree horizontal FoV. This is the motivation why the generated panoramic images holds an aspect ratio of 2:1 instead of 4:1.

2. The authors claim that this is the first work on street scene generation full of various object structures. This might not be true. The following works also address street-view scene generation:

a. Yan, Yunzhi, et al. "Streetcrafter: Street view synthesis with controllable video diffusion models."CVPR 2025.
b. Ze, Xianghui, et al. "Controllable Satellite-to-Street-View Synthesis with Precise Pose Alignment and Zero-Shot Environmental Control." ICLR 2025.
c. Lin, Tao Jun, et al. "Geometry-guided cross-view diffusion for one-to-many cross-view image synthesis." 3DV 2025.

3. The proposed method is of limited novelty. As also stated by the authors, the dual-branch architecture is borrowed from PanFusion, and the CAA is borrowed from MVDiffusion. The differences are that this paper introduces panoramic and multi-view structural prompts, such as segmentation maps and contour maps, and employs a panoramic branch to assist the multi-view branch explicitly. However, the former is just an introduction to additional inputs. The latter is also used in other dual-branch works, using one branch to guide the other branch. For example, PanFusion leverages the multi-view branch to guide the panoramic branch.

4. I do not really get the key differences between this work and PanFusion. From my understanding, the differences include:

a. This paper employs additional structure conditions, such as segmentation maps and contour maps, which are not used in PanFusion. However, I don't think this is a key difference. The structure conditions can also be easily applied to PanFusion by ControlNet or other related methods. This should not be a key difference or significant contribution of this paper.

b. This paper leverages the panoramic branch to assist the multi-view branch, while in PanFusion, the two branches exchange information with each other. PanFusion uses the output from the panoramic branch, while this paper stitches the outputs from the multi-view branch. I understand there are different intuitions behind them. However, I still don't think this is a significant contribution.

**Questions:**

Plz refer to my comments in "Weakness".

Some detailed comments:

1. There are formatting issues in Lines 179-180.
2. In Lines 195-196, the authors state that PanFusion generates panoramas and uses them to supervise multi-view results. However, if my understanding is not wrong, the PanFusion is bijective. The multi-view branch also assists the panoramic branch.

---

> ### Author Response · Authors · 2025-11-26
> **Response to Reviewer dsEf**
>
> ### **Q1: Resolution and aspect ratio issue.**
>
> **A1:**  (1)  Regarding the resolution issue, our current model already generates 2K panoramic images (Panfusion 1k), which is higher than most existing methods. While the output resolution can indeed be further enhanced by super-resolution, our approach directly synthesizes high-quality details during generation. The current resolution is mainly limited by GPU memory (48 GB). With a larger memory capacity, our framework can scale up to 1024×1024 per view, enabling 4K–8K panoramas accordingly.
>
> (2) Our eight-view design corresponds to the four horizontal cube faces but is further split into eight overlapping perspective views. Unlike standard cube-map generation (e.g., CubeDiff), where faces only meet at boundaries, the overlapping regions in our views provide strong multi-view consistency constraints, which significantly reduce cross-view discontinuities. In addition, since our dataset is captured by vehicle-mounted cameras, the top and bottom poles mostly contain non-informative or irregular content (e.g., car roof, vehicle hood). Therefore, we remove these poles and focus the generation on the meaningful horizontal region.
>
> ### **Q2: Related works on street scene generation.**
> **A2:**  We have added descriptions of these works (e.g., StreetCrafter, SatDreamer360, Geometry-guided Cross-view Diffusion) in L148–155. These works indeed advance cross-view and street-view synthesis by leveraging LiDAR/BEV or geometry-guided diffusion to control structure and pose. However, they principally focus on single-view or satellite↔ground conversions and on precise pose/conditioning for video or one-to-many cross-view tasks. In contrast, our work specifically targets multi-view consistent image generation and subsequent stitching into partial panoramas.
>
> ### **Q3, Q4: Novelty and difference from PanFusion.**
> **A3/A4:**  We agree that our framework shares a dual-branch appearance with PanFusion. However, our work introduces a fundamentally different interaction paradigm and cross-space alignment mechanism between the panoramic and multi-view branches.
>
> 1. **Reversed guidance direction with explicit geometric grounding.**
>    In PanFusion, the multi-view branch guides the panoramic branch implicitly through feature fusion. In contrast, we design an opposite guidance flow — the panoramic branch provides global context and scene priors to explicitly guide the multi-view branch. This reversal is not a superficial design choice but a key to ensuring cross-view geometric consistency, since adjacent viewpoints in the multi-view branch correspond to neighboring regions in panoramic space.
>
> 2. **Panorama Alignment Module (PAM): novel cross-space sampling mechanism.**
>    We introduce a PAM that allows points in the multi-view feature space to sample from the neighborhood of their geometrically corresponding locations in the panoramic feature space. This projection-based alignment builds an explicit pixel-to-region correspondence between the two spaces — a mechanism not present in PanFusion or MVDiffusion, which rely on latent feature mixing without geometric projection.
>
> 3. **Staged training for consistent learning.**
>    Our three-stage strategy progressively (i) learns panoramic priors, (ii) enhances intra-view consistency within the multi-view branch, and (iii) jointly aligns both branches for consistency-aware generation. This design leads to stable convergence and smooth cross-view transitions, as shown in our video demos. To verify its necessity, we include an additional ablation: an end-to-end single-stage training variant leads to substantial performance degradation (see Table 7), demonstrating that the staged procedure plays a critical functional role rather than serving as an implementation convenience.
>
> 4. **Structural priors as controllable consistency cues.**
>    The segmentation and contour conditions are not simple extra inputs — they serve as cross-branch structural constraints that ensure spatial correspondence across both panoramic and multi-view domains. These priors play a crucial role in guiding the projection alignment process within PAM, not merely as ControlNet-like additions.
>
> The novelty is not in the components themselves but in how cross-space alignment, guidance flow, and progressive training jointly enable multi-view consistency, which previous dual-branch approaches cannot achieve.
>
> ### **Q5: Formatting error**
> **A5:**  Thank you for pointing out the error. We have corrected the error and updated the paper in Lines 194-195 now.
>
> ### **Q6: PanFusion is bijective.**
> **A6:**  We revised our description of PanFusion (see L207-211 in the updated version) to accurately reflect that, while PanFusion performs bidirectional feature exchange, the final supervision is applied to the panoramic branch using multi-view images. In contrast, our method explicitly reverses this guidance: the panoramic branch provides priors to enhance the multi-view branch.

---

> > ### Author Response · Authors · 2025-11-29
> > **Response to Reviewer dsEf**
> >
> > We appreciate the reviewer’s detailed comments. We have addressed all raised concerns by (i) clarifying the resolution/aspect‐ratio design, (ii) adding and refining the related works, (iii) providing a clearer comparison with PanFusion and explaining our methodological novelty, (iv) correcting the previous formatting issues, and (v) revising our description of PanFusion to accurately reflect its bidirectional feature exchange mechanism. We hope these clarifications and new results help improve your evaluation. If you have any further questions or suggestions, we would be happy to continue refining the paper.

---

### Official Review · Reviewer_TAN1 · 2025-11-01

**Soundness:** 3
**Presentation:** 3
**Contribution:** 3
**Rating:** 6
**Confidence:** 5

**Summary:**

This paper presents StreetDiffusion, a panorama–perspective synergy framework for generating multi-view urban street scenes from text and structural prompts. The method pairs a panoramic branch with several perspective branches and connects them through a Panorama Alignment Module that maps local views onto the sphere to enforce cross-view geometric consistency while incorporating structure priors such as semantic segmentation, contours, or sketches via ControlNet-style adapters and attention. A global-to-local prompt design supplies scene-level context and view-specific detail, and training proceeds in stages to first stabilize each branch and then optimize joint alignment. The authors introduce the Street360 dataset with panoramic images and corresponding multi-view crops, and benchmark against strong baselines including PanFusion and MVDiffusion using FID, IS, CLIP-Score, overlapping-region PSNR, and a user study. Results show higher fidelity and stronger cross-view coherence, particularly when structure conditions are provided, advancing controllable street-scene generation.

**Strengths:**

1. **Elegant and effective framework design**: Structural cues such as segmentation, outline, and sketch are uniformly injected, while panoramic branches and multi-perspective branches are collaboratively modeled, providing each other with context and constraints, and can simultaneously take into account global layout and single-view details.
2. **Robust Structural Control**: is a key advantage of StreetDiffusion, which generates reasonable street scene skeletons even with weakly constrained inputs such as rough sketches or incomplete outlines.
3. **Good writing**: This article has a clear structure and logical flow, making it easy to understand the basic principles.
4. **Extensive and comprehensive experiments**: numerous experiments have demonstrated that StreetDiffusion can effectively generate high-quality and realistic street view images based on various structural cues, and the ablation experiments provide sufficient evidence.

**Weaknesses:**

1. **Insufficient experimental description**: In the comparative experiments, the text prompt input for the comparative methods needs clarification.
2. **The credibility of the user evaluation results is insufficient**: In the User Study Results, the authors only counted 21 valid questionnaires derived from 20 generated images. Is this representative?
3. **Poor intra-image consistency**: Qualitative results (Fig 3) show that although the street view images generated by StreetDiffusion are more realistic and accurate in structure and texture, there seems to be a significant inconsistency in texture style under different viewpoints (for example, different buildings exhibit different texture styles at every 90° angle). The authors need to provide further explanation.

**Questions:**

1. In the comparative experiments, the text input for different methods is ambiguous. StreetDiffusion utilizes GPT4 to describe multi-view images and generate local text when generating text prompts. However, does the input in the baseline also use global + local text as input? Will different text input lengths lead to unfair comparisons?
2. Intra-image consistency is crucial for 360° street view panoramic images, especially the consistency between the two sides of the image. For example, if the generated result is rotated 180° along the camera viewpoint, does the resulting image have obvious inconsistencies (e.g., obvious stitching marks)? I would love to see the authors discuss the above issues.

---

> ### Author Response · Authors · 2025-11-26
> **Response to Reviewer TAN1**
>
> ### **Q1, Q4: Clarify the text prompt input for the comparative methods**
>
> **A1, A4:**  The text prompts used for all comparative methods are identical to those used in our approach, ensuring a fair comparison. The only difference is the way these texts are formatted to match each model’s native input interface. As stated in Lines 181–188, our multi-view branch adopts eight separate prompts, identical to the input format of MVDiffusion, with each sentence corresponding to one camera view. In contrast, panoramic baselines, as well as the panoramic branch of our own model, take a single concatenated prompt formed by merging the same eight sentences. It is important to emphasize that the concatenated prompt contains precisely the same semantic content as the eight separate prompts. The variation lies solely in formatting rather than in the information provided, so no method receives any advantage or additional guidance. To prevent potential ambiguity, we have clarified this point in the revised paper in Lines 188–193.
>
> ### **Q2: The user evaluation results are insufficient**
>
> **A2:**  In our original submission, our user study was based on 20 samples per model under identical prompts, with three evaluation criteria (a total of 60 questions).  To enhance the credibility and representativeness of the results, we have expanded the user study: each model now includes 45 generated samples (135 questions in total), and the number of participants has been increased to 100.  The updated results are provided in Table 2.
>
> | **Method**  | **Style Consist.** | **Realism** | **Multi-view Consist.** |
> | ----------- | ------------------ | ----------- | ----------------------- |
> | MVDiffusion | 16.69%             | 19.84%      | 18.82%                  |
> | PanFusion   | 1.78%              | 2.45%       | 2.88%                   |
> | **Ours**    | **81.53%**         | **77.71%**  | **78.30%**              |
>
> ### **Q3: Poor intra-image consistency in Fig. 3**
>
> **A3:**  The slight texture style inconsistency across different viewpoints mainly arises from the independent noise initialization and view-specific feature differences during multi-view generation.
> Our model already mitigates this issue through the Panorama Alignment Module (PAM), which allows each point in the multi-view space to sample from the neighborhood of its corresponding location in the panoramic space, thereby improving cross-view texture coherence.It is worth emphasizing that such minor inconsistencies are a common limitation shared by existing multi-view or panoramic generation methods (e.g., MVDiffusion, Panfusion, Text2Light), which often exhibit even more visible view-to-view texture shifts in challenging indoor scenes. Compared with these approaches, our method achieves notably better intra-image and cross-view consistency, as validated in both quantitative metrics (see Table 1 and Table 5) and qualitative comparisons (see Figure 3).
> ### **Q5: Consistency of the left–right boundary in 360° panoramic images**
>
> **A5:**  The left–right boundary inconsistency is indeed a long-standing challenge in panoramic image generation. Our method naturally avoids this issue through its multi-view generation framework: although the left and right edges of the panorama are not adjacent in image space, they correspond to neighboring viewpoints in the multi-view domain. The multi-view branch enforces cross-view consistency, ensuring that boundaries align seamlessly.
>
> In addition, the panoramic branch models global 360° scene context and applies circular padding during training to maintain continuity across the 0°–360° seam. Finally, our Panorama Alignment Module (PAM) injects panoramic knowledge into the multi-view branch, further enhancing consistency across long-range spatial regions.
>
> As shown in our video demo, the panoramic results exhibit smooth transitions between neighboring views without visible stitching artifacts. Moreover, as shown in Fig. 3, our model achieves stronger multi-view consistency, further validating the effectiveness of our design.

---

> > ### Author Response · Authors · 2025-11-29
> > **Response to Reviewer TAN1**
> >
> > We hope these additions improve the clarity and credibility of our work. In summary, we have carefully addressed the concerns you raised, including clarifying the text inputs in the comparative experiments, adding analysis on the representativeness of the user study, and expanding the discussion on intra-image consistency. We also explicitly analyzed the consistency of the left–right boundary in 360° panoramic images in disscussions.
> >
> > We hope these additions improve the clarity and completeness of our responses. If you have any further questions or feel that additional clarification would be helpful, we would be glad to continue refining the manuscript.

---

### Official Review · Reviewer_a1Fw · 2025-11-01

**Soundness:** 3
**Presentation:** 3
**Contribution:** 3
**Rating:** 6
**Confidence:** 3

**Summary:**

This paper introduces StreetDiffusion, a novel multi-view stable diffusion model designed to generate realistic and consistent urban street scenes, a task more complex than indoor or wild scene generation. To support this, the authors created Street360, a new large-scale dataset of 10,000 multi-view panoramic street images with corresponding structural annotations. The StreetDiffusion model employs a Panorama-Perspective Synergy Framework to integrate global panoramic information with local perspective views, using structure prompts (like segmentation maps, contour maps, or user sketches) to guide the generation process and ensure structural accuracy. A Panorama Alignment Module (PAM) is also introduced to enforce geometric consistency between the panoramic and perspective representations. Experiments show that StreetDiffusion outperforms existing multi-view generation methods, producing higher-quality and more consistent street scene images that adhere to the provided structural cues.

**Strengths:**

The paper introduces Street360, the first large-scale, high-resolution dataset specifically designed for the complex and under-studied task of multi-view urban street scene generation, complete with valuable structural annotations.

It proposes a novel model, StreetDiffusion, which uses a Panorama-Perspective Synergy Framework to uniquely integrate global panoramic information with local multi-view synthesis, guided by structural prompts like segmentation maps or sketches for enhanced control and realism.

The inclusion of a Panorama Alignment Module (PAM) ensures geometric consistency between views, allowing the model to demonstrably outperform existing methods in generating high-quality, realistic, and structurally coherent street panoramas.

**Weaknesses:**

1. The model's optimal performance relies heavily on the availability and quality of detailed structure prompts (like segmentation or contour maps). While the paper demonstrates results using only text and user sketches, the quality and consistency might be reduced without this strong structural guidance.

2.The model employs a complex three-stage training strategy. This multi-stage approach can be more cumbersome and computationally expensive than a single, end-to-end training process.

3.The panoramic module synthesizes at $2048 \times 512$ resolution, while the multi-view branch trains on $512 \times 512$ images. This means the final stitched panorama (approximately $2048 \times 512$) is a significantly lower resolution than the 4K to 8K images available in their own Street360 dataset, indicating a gap in the model's ability to generate truly high-resolution content.

4.The method is, to a large extent, a clever combination of existing technologies. It borrows the dual-branch architecture from PanFusion and the CAA from MVDiffusion , and it integrates standard components like ControlNet and LORA.

5.The paper includes an extra experiment on the Matterport3D indoor dataset (Table 7). The results show that while StreetDiffusion achieves the best FID score, it performs worse than the baseline MVDiffusion on the IS and CS metrics. This suggests that the model's architecture, while effective for street scenes, is not universally applicable and may be less effective for indoor environments.

**Questions:**

Given the authors' insightful future work direction on exploring BEV representations, it might be valuable to also discuss or reference recent related works, such as CrossViewDiff and SkyDiffusion, as these could serve as relevant points of comparison for that line of inquiry.

---

> ### Author Response · Authors · 2025-11-26
> **Response to Reviewer a1Fw**
>
> ### **Q1: The quality and consistency might be reduced without this strong structural guidance**
>
> **A1:**  In fact, when strong structural conditions (e.g., semantic segmentation maps) are provided, the consistency of the generated images is slightly better than with weak conditions (e.g., sketches), as the former inherently contains richer structural information. However, the gap is not significant—due to our panoramic module, which ensures multi-view consistency at the global level, a capability that existing methods lack. Moreover, weak conditions are easier to obtain in real-world applications. Nevertheless, as shown in Table 1 and Table 5 (in the revised paper), our model consistently outperforms previous methods under the same input conditions (text+seg, text+contour, or only text).
>
> ### **Q2: The multi-stage approach can be more cumbersome and computationally expensive**
>
> **A2:**  While our three-stage training strategy may seem more complex than a single end-to-end pipeline, it is essential for achieving stable convergence and strong multi-view consistency. Specifically, in Stage 1, we pretrain both the multi-view and panorama branches to learn general street-scene and panoramic priors, which provide crucial geometric and appearance knowledge. Stage 2 focuses on enhancing cross-view consistency within the multi-view branch. Finally, Stage 3 performs joint training to inject the panoramic priors into the multi-view branch, further improving view alignment and coherence.
>
> We conducted an ablation study on the Street360 dataset with text-only prompts:
>
> | Method         | FID ↓     | IS ↑     | CS ↑      | OP_PSNR ↑ |
> | -------------- | --------- | -------- | --------- | --------- |
> | end-to-end     | 22.35     | 4.98     | 22.25     | 38.71     |
> | 3-stage (Ours) | **11.95** | **6.02** | **24.93** | **39.12** |
>
> We empirically found that direct end-to-end training leads to unstable optimization and severe cross-view inconsistency, while the staged approach yields more robust and visually coherent results. These results have been added to Table 6 (Page 10).
>
> ### **Q3: Lower resolution than the 4K to 8K images in the dataset**
>
> **A3:**  We would like to clarify that our current panoramic synthesis (2048×512) already offers higher resolution than existing methods, which typically generate panoramas at around 1K resolution. The current resolution choice is mainly constrained by GPU memory (A6000 48G) and training cost. Importantly, our framework scales naturally with the training resolution--for example, when the multi-view branch is trained at 1024×1024, the stitched panorama can reach approximately 4K resolution, and higher resolutions are attainable with more resources. Nevertheless, our goal in this paper is to demonstrate the effectiveness of our multi-view consistent generation framework, and we view high-resolution synthesis as an orthogonal extension.
> ### **Q4: The method is, to a large extent, a clever combination of existing technologies**
>
> **A4:**  Our method is driven by the goal of achieving geometrically consistent multi-view generation, and our main contributions focus on enabling explicit cross-space alignment rather than assembling existing components. The core novelty of our framework lies in how panoramic information is injected into the multi-view generation process in a geometry-aware and structurally consistent manner. Specifically, our approach projects points from the multi-view space into the panoramic space, and the Panorama Alignment Module (PAM) enables sampling within the local neighborhood of the corresponding panoramic points. This allows the multi-view branch to leverage panoramic priors more effectively, enhancing cross-view coherence. Although dual-branch architectures and cross-attention mechanisms have appeared in prior works (e.g., PanFusion, MVDiffusion), our projection-based neighborhood sampling combined with staged training is novel and crucial for aligning multi-view and panoramic features. Besides, modules like ControlNet and LoRA are included solely for training stability and efficiency.

---

> ### Author Response · Authors · 2025-11-26
> **Response to Reviewer a1Fw (2)**
>
> ### **Q5: Worse results on the Matterport3D indoor dataset**
>
> **A5:**  Our method is primarily designed for complex outdoor street scenes, where it leverages street-specific priors in both multi-view and panoramic branches. Indoor environments have been well studied in prior work, and our framework was not fine-tuned on Matterport3D, so the reported results reflect a zero-shot evaluation rather than the intended use case.
>
> In Table 8 (in the Appendix of the revised version), we provide updated metrics based on indoor-trained models for a fairer comparison:
>
> | Method                       | FID ↓     | IS ↑     | CS ↑      |
> | ---------------------------- | --------- | -------- | --------- |
> | Text2Light                   | 43.66     | 4.92     | 25.88     |
> | SD+Lora                      | 23.02     | 6.58     | 28.60     |
> | PanFusion                    | 19.88     | 6.50     | 24.98     |
> | MVDiffusion                  | 21.44     | 7.32     | **30.04** |
> | StreetDiffusion (Zero-shot)  | 16.26     | 6.31     | 24.51     |
> | StreetDiffusion (Fine-tuned) | **13.16** | **7.38** | 29.38     |
>
> Importantly, our core contribution remains in outdoor multi-view consistent generation, where the model achieves state-of-the-art performance.
>
> ### **Q6: Discuss related works such as CrossViewDiff and SkyDiffusion**
>
> **A6:**  We sincerely thank you for the helpful suggestion. We agree that recent works such as CrossViewDiff and SkyDiffusion are related to our future direction on BEV representations. Both approaches utilize diffusion-based frameworks for cross-view generation, particularly focusing on perspective-to-BEV or sky-view translation. In contrast, our current work aims to generate multi-view consistent images from input views, which can later be stitched into a partial panorama (not a full 360° panorama, as the top and bottom poles are excluded). Our focus is thus on maintaining geometric and visual consistency across overlapping views, rather than directly learning BEV or top-down representations. We appreciate the reviewer’s insightful comment and will include a discussion of these works and their relation to our approach in the revised version.We have added the corresponding citations and discussion in Lines 116–119 of the revised paper.

---

> ### Comment · Reviewer_a1Fw · 2025-11-27
>
> My questions have been largely answered; However, I feel that the placement of your supplementary reference on line 116 is not quite correct. Perhaps it would be more appropriate in the "Outdoor Generation" section below.

---

> ### Author Response · Authors · 2025-11-27
> **Response to Reviewer a1Fw**
>
> Thank you for your feedback and for raising your rating to accept. We appreciate your suggestion regarding the placement of the supplementary reference on line 116. We have moved it to the “Outdoor Generation” section as recommended.（Lines 146-149）

---

### Author Response · Authors · 2025-12-03
**Final Comment for Paper #10723**

Dear Area Chair and Reviewers,

We sincerely thank you for your thoughtful feedback and valuable suggestions, which have greatly helped us improve the quality of our manuscript.

We would like to highlight that we have carefully addressed all concerns raised by the reviewers in our revised submission. These revisions have significantly enhanced the paper’s clarity, methodological rigor, and overall contribution to the field. In particular, we are grateful that Reviewer a1Fw explicitly acknowledged the improvements and has updated their assessment to recommend acceptance.

We fully understand the constraints imposed by the current discussion policy, which limits further direct interaction with reviewers during the rebuttal phase. Nonetheless, we hope that the revised version now meets the high standards expected for publication at ICLR 2026.

Thank you once again for your time, careful consideration, and constructive engagement with our work.

Best regards,
The Authors of Paper #10723

---

### Meta-Review · Area_Chair_jkyy · 2026-01-07

**Summary:**

The paper proposes StreetDiffusion, a multi-view diffusion framework for generating street scene panoramas using a dual-branch architecture (panoramic and perspective) with a Panorama Alignment Module (PAM). The authors also introduce a dataset of 10K street scenes with structural annotations.

While the reviewers acknowledge the usefulness of the proposed Street360 dataset and the authors' efforts during the rebuttal phase (specifically expanding the user study and clarifying text prompt fairness), the consensus leans towards rejection. The decision is primarily driven by concerns regarding limited novelty and excessive engineering complexity.

The most critical concern, shared by multiple reviewers, is that the proposed method appears to be an incremental engineering adaptation of existing works, particularly PanFusion. And some reviewers also mentioned the proposed method being too complex relative to the problem it solves.

Based on above, I think the contribution of this work does not meet the high bar for acceptance at ICLR.

**Reviewer Concerns:**

Many of the concerns have been addressed, including indoor/zero-shot performance (expand study to 100 participants), the fairness in comparison, and training complexity, also the presentation errors. However, the primary concerns that the method is an incremental engineering adaptation of PanFusion remain unresolved, and the weak task motivation (Ynee) remains unconvinced from my point of view.

**Reviewer Scores:**

**a1Fw**: likely maintain the 6, as they explicitly stated their concerns were largely answered by the rebuttal.

**TAN1**: They would likely maintain the 6 score, but could also increase 8, given that the authors resolved their primary complaint by expanding the user study size.

**dsEf**: they would likely maintain the 4, as I think they remained unconvinced regarding the novelty compared to PanFusion despite the authors' defense.

**Ynee**: they might keep the 2 or improve slightly to 4, but likely remain since the reviewer fundamentally disagreed with the motivation of the task itself.

---

### Decision · Program_Chairs · 2026-01-26

Reject